# Learning-Augmented Scalable Linear Assignment Problem Optimization via Neural Dual Warm-Starts

Ilay Yavlovich [1] [*]   Jad Agbaria [1] [*]   Muhamed Mhamed [1]   Nir Weinberger [1]   Jose Yallouz [1]

## Abstract

The Linear Assignment Problem is a fundamental combinatorial optimization task where classical exact solvers ensure optimality but suffer from an $\mathcal{O}(N^3)$ bottleneck, while recent neural approximations struggle with scalability and exactness. We propose a *learning-augmented* framework that accelerates exact solvers by predicting dual variables to warm-start the search, backed by a fallback mechanism to preserve worst-case guarantees. Central to our approach is **RowDualNet**, a lightweight, row-independent architecture that avoids the $\mathcal{O}(N^2)$ memory bottleneck of graph models, enabling scalable neural warm-starting up to $N = 16{,}384$. Feasibility is guaranteed by construction via the Min-Trick mechanism, completely eliminating the need for costly iterative projections. Empirically, our method drastically reduces the search effort of the Jonker-Volgenant (LAPJV) algorithm, yielding robust zero-shot generalization with strict optimality and end-to-end speedups of over 2x on complex synthetic data, 1.25x on real-world tracking, and 1.5x on transportation networks.

## 1. Introduction

The Linear Assignment Problem (LAP) is a cornerstone of combinatorial optimization, serving as a critical primitive in diverse high-throughput applications, e.g., multi-object tracking in computer vision, resource scheduling in logistics, and facility allocation in operations research. Formally, given a cost matrix $C \in \mathbb{R}^{N \times N}$, the goal is to find a one-to-one matching between rows and columns that minimizes the total cost.

As these application domains continue to scale, the demand for solvers that are both *fast* and *exact* has grown. However, current approaches force a compromise. Classical exact algorithms, such as the Hungarian method (Kuhn, 1955; Munkres, 1957) and its successor, the Jonker-Volgenant (LAPJV) algorithm (Jonker & Volgenant, 1986; 1987), guarantee optimality, but suffer from cubic time complexity $\mathcal{O}(N^3)$. For large-scale instances ($N \geq 10^3$), this computational latency becomes an issue for real-time applications.

On the other hand, recent neural solvers attempt to replace combinatorial algorithms entirely with deep learning models, e.g Graph Neural Networks (GNNs) (Liu et al., 2024; Aironi et al., 2024). While these methods offer rapid inference, they do not eliminate the computational bottleneck. First, they are *approximations*, often treating hard constraints as soft loss terms or rely on greedy post-processing, failing to guarantee valid or optimal solutions. Moreover, they struggle with *scalability* due to quadratic memory complexity. Architectures relying on edge convolutions typically require $\mathcal{O}(N^2 \cdot H)$ memory to maintain high-dimensional latent representations for every edge, causing memory exhaustion on instances larger than $N \approx 2{,}000$ (where hidden dimensions $H$ inflate the memory footprint by orders of magnitude). Thus, while they ameliorate the runtime issues of $\mathcal{O}(N^3)$ classical algorithms, they cannot scale to the massive instances required by modern applications.

In this paper, we propose an intermediate path: using deep learning not to replace the solver, but to *accelerate* it. We leverage the theory of primal-dual optimization, which states that initializing an exact solver with dual variables close to the optimum can drastically reduce the search space. While this "learned duals" model has been explored theoretically (Dinitz et al., 2021), prior attempts have relied on expensive iterative algorithms to project invalid predictions onto a feasible set, negating the speed advantage of the neural network, and do not scale up to larger instances.

**Summary of Contributions.** We propose **Neural Dual Warm-Starting**, a learning-augmented framework that combines the inference speed of deep learning with the optimality guarantees of exact algorithms. To the best of our knowledge, this is the first approach that scales to industrial problem sizes ($N = 16{,}384$) while preserving worst-case

---

[*]Equal contribution   [1]Department of Electrical and Computer Engineering, Technion – Israel Institute of Technology, Haifa, Israel. Correspondence to: Ilay Yavlovich <ilayy@campus.technion.ac.il>.

*Proceedings of the 43rd International Conference on Machine Learning*, Seoul, South Korea. PMLR 306, 2026. Copyright 2026 by the author(s).

safety. Our main contributions are:

- **Neural Dual Warm-Starting Framework:** We introduce a solver-aware framework that predicts dual variables to warm-start exact solvers. By utilizing a constructive **Min-Trick mechanism**, we guarantee dual feasibility by definition, effectively bypassing the expensive initial phases of the combinatorial search while ensuring fallback to cold-start execution with no asymptotic degradation.
- **RowDualNet Architecture:** We propose RowDualNet, a row-independent neural architecture that avoids the $\mathcal{O}(N^2)$ memory bottleneck of GNNs, enabling neural warm-starting at scale up to $N = 16{,}384$.
- **Exact and Scalable Performance Gains:** We illustrate that the implementation of neural dual warm-starting results in an enhancement of over **2×** **end-to-end speedup** when applied to complex synthetic distributions; furthermore, it demonstrates improvements of over **1.25×** and **1.5×** **end-to-end speedup** on empirical datasets pertaining to tracking and transportation, respectively, when compared to cold-start LAPJV baselines, while simultaneously ensuring **exact optimality** and maintaining robustness in the context of distributional shifts.

## 2. Preliminaries

We consider the LAP for a cost matrix $C \in \mathbb{R}^{N \times N}$, where $C_{ij}$ represents the cost of assigning agent $i$ to task $j$. The primal optimization problem seeks a permutation matrix $X \in \{0,1\}^{N \times N}$ that minimizes the total cost:

$$\min_X \sum_{i,j} C_{ij} X_{ij} \quad \text{s.t.} \quad \sum_j X_{ij} = 1 \; \forall i,$$
$$\sum_i X_{ij} = 1 \; \forall j, \; X_{ij} \in \{0,1\}. \tag{1}$$

Because the constraint matrix of this Integer Linear Program (ILP) is totally unimodular, its continuous Linear Program (LP) relaxation is guaranteed to admit an integral optimal solution (Hoffman & Kruskal, 1956). The corresponding dual problem seeks to maximize $\sum_i u_i + \sum_j v_j$ subject to the feasibility constraint $u_i + v_j \leq C_{ij}$ for all pairs $(i,j)$, where $u \in \mathbb{R}^N$ and $v \in \mathbb{R}^N$ are the row and column potentials, respectively. While solving the continuous LP is theoretically sufficient, specialized combinatorial algorithms vastly outperform general-purpose LP solvers by exploiting this primal-dual structure.

**Primal-Dual Optimality.** The complementary slackness condition dictates that a feasible assignment $X$ and feasible duals $(u,v)$ are optimal if and only if $X_{ij} = 1 \implies u_i + v_j = C_{ij}$. We define the *reduced cost* as $r_{ij} =$

$C_{ij} - u_i - v_j \geq 0$. The edges where $r_{ij} = 0$ form the *equality subgraph*. Classical dual solvers operate by iteratively updating $u$ and $v$ until a perfect matching can be found entirely within this equality subgraph.

**The LAPJV Algorithm.** The Jonker-Volgenant (LAPJV) algorithm (Jonker & Volgenant, 1986; 1987) remains the gold standard for exact linear assignment. It operates in two phases:

1. **Greedy Match Step (Initialization):** Fast, static heuristics (e.g., column reduction) are applied to rapidly construct an initial set of feasible duals and a partial primal matching.

2. **Augmentation Phase:** For any row left unassigned by the greedy step, the algorithm executes a shortest-augmenting-path search on the residual graph. This search dynamically updates the dual potentials to expose new zero-reduced-cost edges until the assignment can be successfully augmented.

While the greedy initialization is highly efficient ($\mathcal{O}(N^2)$), its static heuristics ignore complex, distribution-specific structures. When this phase fails to resolve a large portion of assignments, the algorithm is forced into a lengthy augmentation phase. Because each of the $\mathcal{O}(N)$ unassigned rows may require a shortest-path search over $\mathcal{O}(N^2)$ edges, this phase dictates the worst-case cubic complexity $\mathcal{O}(N^3)$ and serves as the primary computational bottleneck of the solver.

## 3. Method: Neural Dual Warm-Starting

We propose a learning-augmented framework designed to accelerate the LAP solving process without sacrificing optimality. Our method operates on the principle of *dual warm-starting*: instead of asking a neural network to output the discrete assignment matrix $X$ (which is prone to invalidity), the network is trained to predict the continuous row potentials $\hat{u}$. We then constructively derive the column potentials $\hat{v}$ to guarantee dual feasibility, providing the exact solver with an equality subgraph that drastically prunes the search space. Given a cost matrix $C \in \mathbb{R}^{N \times N}$, our end-to-end pipeline (Fig. 1) consists of four sequential stages:

### 3.1. Row-Centric Feature Extraction

We first compress the cost matrix into a lightweight feature matrix $F \in \mathbb{R}^{N \times D}$. We select a compact set of $D \ll N$ features, detailed in App. B, designed to capture critical signals such as: **cost distribution** (e.g., min, mean), **ambiguity** (e.g., entropy), and **global competitiveness** (e.g., rank statistics). This specific feature set was chosen to provide a sufficient summary of an agent's "market power" while

*Figure 1.* **Pipeline for Neural Dual Warm-Starts.** We extract row-centric features, predict row potentials $\hat{u}$ via RowDualNet, and construct feasible column potentials $\hat{v}$ using the Min-Trick. This valid dual pair seeds the LAPJV solver. A fallback mechanism (red path) ensures robustness by reverting to a cold start if the prediction quality is low.

maintaining $\mathcal{O}(N \cdot D)$ complexity, enabling scalability to massive instances.

### 3.2. Neural Prediction (RowDualNet)

The feature matrix $F$ is passed through **RowDualNet**, a row-independent deep neural network that treats the problem as a set-processing task (further detailed in App. B). To capture inter-agent competition efficiently, it employs a two-stage mechanism:

1. **Independent Encoding:** First, a shared MLP transforms each row's features into an initial dual estimate $\hat{u}_{init}$.
2. **Reduced Cost k-NN Refinement:** We then apply a lightweight "sparse refinement" step. For each row $i$, we identify the $k$ columns with the smallest "pseudo-reduced costs" ($C_{ij} - \hat{u}_{init,i}$) and aggregate their corresponding values. This operation can be interpreted as a $k$-Nearest Neighbor search in the dual space, informing the agent of its "best" options' prices relative to its current bid. Because the optimal assignment is sparse (each agent matches exactly one task), the relevant competitive information is highly concentrated in the few columns with near-zero reduced costs. Thus, a small fixed $k \ll N$ is sufficient to capture the necessary signal even for massive $N$.

The final output is the scalar potential $\hat{u}_i$. We employ a supervised learning approach where the ground-truth dual variables $u^*$ are generated offline using the exact LAPJV solver. The training objective minimizes the Mean Absolute Error (MAE) between $\hat{u}$ and $u^*$, regularized by a **Complementary Slackness Loss** that enforces tightness on ground-truth optimal edges. This architecture ensures that memory usage scales linearly with $N$, avoiding the bottlenecks of prior full-graph approaches.

### 3.3. Feasibility via the Min-Trick.

Given the predicted row potentials $\hat{u}$, we compute the corresponding column potentials $\hat{v}$ using the *Min-Trick*: $\hat{v}_j = \min_i(C_{ij} - \hat{u}_i)$. This operation guarantees that the resulting pair $(\hat{u}, \hat{v})$ satisfies the dual feasibility constraint $\hat{u}_i + \hat{v}_j \leq C_{ij}$ for all $i, j$ by construction. This step is

relatively computationally efficient ($\mathcal{O}(N^2)$ but highly parallelizable) and eliminates the need for iterative projection algorithms.

### 3.4. Exact Resolution (Seeded LAPJV).

Finally, the feasible duals $(\hat{u}, \hat{v})$ and the cost matrix $C$ are passed to the LAPJV solver, which we implemented by utilizing a custom wrapper around the optimized LAP library core that exposes the dual initialization arrays. Instead of the standard heuristic initialization (which starts with row minima), we inject our predicted $\hat{u}$ and $\hat{v}$ directly into the solver's memory structures before the first reduction phase. This initializes the solver's internal state with a dense equality subgraph ($r_{ij} = C_{ij} - \hat{u}_i - \hat{v}_j \approx 0$), allowing it to bypass the computationally expensive "price wars" of the early augmentation phases and converge rapidly to the exact optimal assignment $X^*$.

### 3.5. The Fallback Mechanism

Neural networks, while powerful, can occasionally produce poor predictions. A "bad seed" is a set of potentials $(\hat{u}, \hat{v})$ that results in a sparse or disconnected equality subgraph $E_0 = \{(i,j) : C_{ij} - \hat{u}_i - \hat{v}_j = 0\}$. Initializing the LAPJV solver with such a seed could theoretically degrade performance by forcing the algorithm to perform excessive augmentation steps to repair the duals.

To ensure robustness, we implement a lightweight quality check before invoking the exact solver. We define the *average degree* $\rho$ of the equality subgraph as:

$$\rho = \frac{1}{N} \sum_{i,j} \mathbb{I}(|C_{ij} - \hat{u}_i - \hat{v}_j| < \epsilon) \tag{2}$$

where $\epsilon$ is a numerical tolerance for floating-point comparisons. If $\rho$ falls below a validation-tuned threshold $\tau$, the system flags the prediction as unreliable. In such cases, the neural predictions are discarded, and the LAPJV solver is initialized with standard cold-start heuristics.

**Asymptotic Safety.** This mechanism ensures that the neural overhead does not impact the asymptotic complexity of the solver. The running time overhead $T_{\text{overhead}}$, as detailed in App. A, scales as $\mathcal{O}(N^2 \log(N))$. Since the exact solver

running-time $T_{\text{cold}}$ has a worst-case complexity of $\mathcal{O}(N^3)$ regardless of the specific initialization of the dual variables, we have:

$$\lim_{N \to \infty} \frac{T_{\text{overhead}}}{T_{\text{cold}}} = 0 \qquad (3)$$

Thus, for sufficiently large instances, the cost of the safety check is negligible. Even when the fallback is triggered 100% of the time or if the fallback mechanism fails to detect any bad seed, the total system runtime asymptotically converges to the baseline solver's performance.

## 4. Methodological Guarantees

In this section, we formalize the correctness of our framework and analyze the computational complexity of the inference pipeline compared to classical baselines. Unlike heuristic "neural solvers" that predict assignments directly (often violating constraints), our approach is designed to maintain the invariants of exact optimization.

### 4.1. Optimality by Construction

A primary concern of many neural solvers is their lack of guaranteed optimality. To ensure our method never degrades into an approximation, we leverage the *Min-Trick* mechanism to enforce dual feasibility by design.

**Proposition 1. (Exactness Preservation).** For any arbitrary output $\hat{u} \in \mathbb{R}^N$ from RowDualNet, the warm-start seed $(\hat{u}, \hat{v})$ generated by the Min-Trick guarantees that the final assignment $X^*$ returned by the LAPJV solver is globally optimal.

*Proof.* The Min-Trick explicitly sets column potentials as $\hat{v}_j = \min_i(C_{ij} - \hat{u}_i)$. By construction, this satisfies $\hat{u}_i + \hat{v}_j \leq C_{ij}$ for all entries, ensuring the seed is *Dual Feasible*. Since LAPJV is a dual-ascent algorithm, the algorithm requires only a feasible dual solution to proceed. Consequently, initializing the solver with any valid dual potentials allows it to continue with the augmenting path phases exactly as it would after reaching such a state through its own internal computations. In this sense, the initialization is mathematically equivalent to resuming execution from a valid intermediate state. The solver proceeds via shortest-path augmentations until Complementary Slackness is satisfied. By the Strong Duality theorem of Linear Programming, the resulting assignment $X^*$ is guaranteed to be optimal. Therefore, the neural network influences only the *convergence speed*, not the *correctness* of the solution. □

### 4.2. Time Complexity

The cold-start LAPJV algorithm has worst-case time complexity $\mathcal{O}(N^3)$. Our neural warm-start introduces an addi-

tional preprocessing overhead that is at most $T_{overhead} = \mathcal{O}(N^2 \log(N))$, consisting of a single pass over the cost matrix to construct dual potentials and evaluate the fallback criterion. As a result, the worst-case asymptotic complexity of the overall pipeline remains unchanged at $\mathcal{O}(N^3)$.

In practice, the warm-start reduces the number of augmentations required by the solver, leading to significant constant-factor speedups, as observed in Sec. 5. A detailed breakdown of the preprocessing costs is provided in App. A.

## 5. Experiments

We evaluated our framework on synthetic datasets designed to challenge combinatorial solvers, as well as real-world tracking and transportation datasets (Shuai et al., 2022; OpenStreetMap, 2025). We focus primarily on three performance dimensions: **End-to-End Runtime** (including all neural overhead), **Stability** (runtime variance), and **Robustness** (generalization to out-of-distribution instances).

Throughout all experiments, we also empirically verified the **Optimality Gap**. As guaranteed by our design, the gap remained exactly $0\%$ across all test instances evaluated, confirming that the neural warm-start strictly preserves the exactness of the solver.

**Hardware and Execution Model.** All experiments were conducted on a single hybrid CPU–GPU node equipped with an *NVIDIA GeForce RTX 3090* and a *12th Gen Intel Core i7-12700*. Classical baselines (`SciPy` and `LAP`) are executed entirely on the CPU. Our method is a hybrid system: the neural warm-start (feature extraction and RowDualNet inference) is executed on the GPU, while the exact LAPJV solver runs on the CPU.

To ensure a fair comparison, we report end-to-end wall-clock runtime, including all components of the pipeline: data transfer between CPU and GPU, feature extraction, neural inference, Min-Trick computation, and execution of the exact solver. No component is excluded or amortized.

To prevent out-of-memory (OOM) errors on massive graphs, all PyTorch reductions for the Min-Trick are processed in chunked $B \times N$ blocks, allowing scalable execution well within standard GPU VRAM limits.

### 5.1. Experimental Setup

**Synthetic Datasets.** We utilized two distinct cost matrix distributions to test robustness:

- **Dense (Uniform) Model:** $C_{ij} \sim U(0, 1)$. A standard benchmark for assignment algorithms. It is considered "hard" for sparse heuristics because the cost structure is unstructured and fully dense, requiring the solver to explore many augmenting paths.

- **Block-Structured Model:** Adapted from Dinitz et al. (2021), this dataset simulates real-world scenarios where agents and tasks belong to specific classes (e.g., doctors are efficient at medical tasks but inefficient at engineering tasks). We generated $L$ groups of agents and tasks ($L = \lfloor N/10 \rfloor$). Base costs between groups are drawn from a skewed discrete distribution, and specific instance costs are perturbed by Gaussian noise, creating a block-diagonal structure.

**Training Protocol.** We employed a Multi-Scale Training to encourage the learning of size-invariant feature representations. We trained a single RowDualNet model on over 1,700 synthetically constructed cost matrices consisting of various sizes $N \times N$ with $N \in \{512, 1536, 2048, 3072\}$ sampled from the synthetic datasets. We then evaluated this single model on instances up to $N = 16,384$ to test out-of-distribution scalability.

**Baselines.** To ensure a rigorous evaluation, we compared our method against two standard exact solver implementations:

- **SciPy:** The `linear_sum_assignment` function from the `scipy.optimize` library (Virtanen et al., 2020), which implements a modified Jonker-Volgenant algorithm with heuristic initialization.
- **LAP:** The optimized $C++$ implementation, `LAP` library (Kazmar, 2021), a state-of-the-art open-source solver for dense linear assignment.

We compared them both against our method evaluating how their relative performance varies depending on problem size and sparsity. By reporting speedups relative to both baselines in each experiment, we ensure that our results reflect true state-of-the-art improvements rather than artifacts of a specific implementation.

**Evaluation Protocol.** Our goal is to measure whether neural dual warm-starting reduces the total time required to solve the assignment problem exactly. Accordingly, we compared our method against standard cold-start solvers as complete systems, rather than attempting to isolate solver-only runtime. This reflects the intended deployment setting, where a lightweight learned component accelerates a classical exact algorithm without compromising optimality. For full details on the model architecture, training curriculum, and hyperparameters, see App. C.

## 5.2. Main Results: End-to-End Runtime

### 5.2.1. SYNTHETIC COST MATRICES

To ensure robust evaluation, we benchmarked performance across problem sizes ranging from $N = 512$ to $N = 16,384$. For each size and distribution (Dense and Block-Structured), we generated 10 independent random cost matrices. We

measured the end-to-end wall-clock time for all methods on these identical instances.

The reported metric is the Mean of Ratios: for each instance $i$, we calculated the individual speedup $S_i = T_{baseline}^{(i)}/T_{ours}^{(i)}$ and reported the arithmetic mean across the 10 trials, preventing outliers in absolute runtime from skewing the results, accurately reflecting the expected time savings per job. The shaded regions in the plots represent the 95% Confidence Interval (CI), illustrating the stability of our method compared to the variance of the baselines. Fig. 2 illustrates the speedup on the Dense Model dataset, while Fig. 3 presents the results for the Block-Structured Model.

**Analysis.** Our method achieves a consistent end-to-end speedup factor $> 1.0$ for all instances with $N \geq 1,024$. For smaller sizes ($N = 512$), the fixed overhead of neural inference constitutes a larger fraction of the runtime. However, the performance gap widens as problem size increases. On the **Dense Model** (Fig. 2), we achieve an average speedup of $\approx 2.0\times$ against `SciPy` and $\approx 2.5\times$ against `LAP` at $N = 16,384$. The gains are even more pronounced on the **Block-Structured Model** (Fig. 3), where the solver effectively exploits the group structure to reach a mean speedup of $\approx 2.25\times$ against `SciPy` and nearly $\approx 4.0\times$ against the `LAP` baseline. Although the large confidence intervals indicate significant variance in the baseline's performance, the robust median speedup ($\approx 2.4\times$) confirms that our method consistently outperforms the baseline in the majority of trials. This variance is driven by the instability of classical solvers on structured inputs, a phenomenon we analyze further in the stability analysis (Sec. 5.4).

### 5.2.2. REAL-WORLD GENERALIZATION: URBAN TRANSPORTATION NETWORKS

The LAP is a core primitive in traffic engineering and logistics, specifically within the **Transportation Problem** (Hitchcock, 1941), which minimizes the cost of distributing commodities from supply sources to destinations. When supply and demand are balanced, this reduces to a bipartite matching problem on the underlying road network. Accelerating these computations is central for logistics operations and dynamic routing in supply chains.

To validate our approach, we constructed cost matrices using the OpenStreetMap (OSM) dataset (OpenStreetMap, 2025). For seven major metropolises (Beijing, New York, Tokyo, Shenzhen, Berlin, Toronto, and Rome) we randomly sampled $N = 10,000$ locations and computed the pairwise shortest-path distances between them. This yields $N \times N$ cost matrices that reflect the specific transportation cost of urban infrastructure. We evaluated performance across 10 random instances per city to ensure statistical significance. Crucially, we tested our pre-trained RowDualNet model

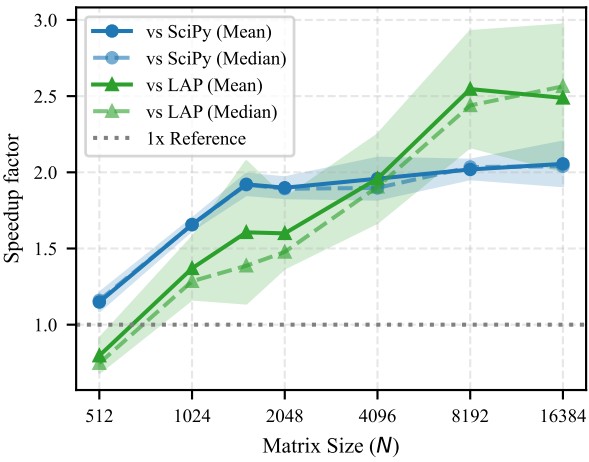

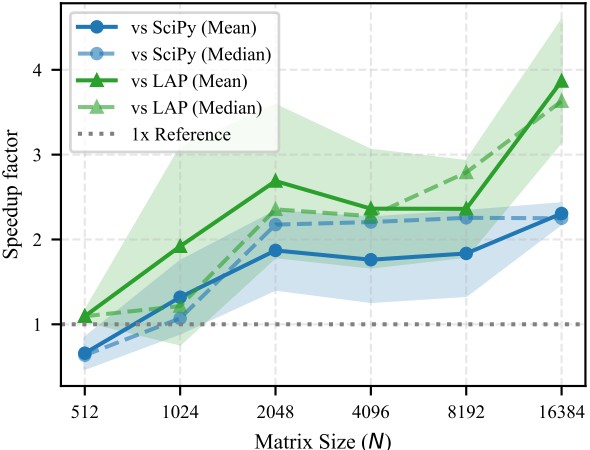

*Figure 2.* **Scalability on Dense Model.** End-to-end speedup factor ($T_{baseline}/T_{ours}$) versus matrix size $N$ on dense matrices (Mean of Ratios $\pm$ 95% CI). The solid lines indicate the mean speedup, while the shaded regions represent the 95% CI. Our Neural Warm-Start framework consistently outperforms both Cold-Start LAP and SciPy when $N \geq 1{,}024$, reaching a peak speedup of $\approx 2.5\times$. The widening shaded area for LAP (green) reflects the high variance of the baseline solver compared to the stability of our method.

*Figure 3.* **Scalability on Block-Structured Model.** End-to-end speedup factor ($T_{baseline}/T_{ours}$) versus matrix size $N$ on Block-Structured Model (Mean of Ratios $\pm$ 95% CI). We achieve substantial acceleration, reaching a mean speedup of $\approx 2.25\times$ against SciPy (blue) and nearly $\approx 4.0\times$ against LAP (green) on the largest instances. The large spread in the CI's (particularly for LAP) highlights the extreme volatility of cold-start baselines on structured data, which our method effectively stabilizes.

(trained only on synthetic data) directly on these matrices *without fine-tuning*.

**Analysis.** Our method generalizes well to the evaluated networks for large datasets. In Fig. 4, we observe a consistent speedup of $\approx 1.4\times - 1.6\times$ relative to SciPy's solver and a major speedup $\approx 1.3\times - 1.8\times$ relative to the LAP solver across all cities. Notably, these gains are sustained despite the specific spatial correlations of urban graphs, which differ significantly from our synthetic training data. Acceleration in diverse scenarios underscores that RowDualNet captures universal structural properties of assignment costs rather than overfitting to idealized distributions, offering a valuable performance advantage for large logistics optimization.

### 5.2.3. REAL-WORLD GENERALIZATION: MULTI-OBJECT TRACKING

We evaluated performance on an additional application of cost matrices derived from the Large Scale Real-World Multi-Person Tracking dataset (Shuai et al., 2022). In Multi-Object Tracking (MOT), the association of detected objects across video frames is formulated as a LAP, where the cost $C_{ij}$ represents the distance $1 - \text{IoU}$ (Intersection over Union) between a tracklet in frame $t$ and a detection in frame $t + 1$. We extracted cost matrices from the dataset with sizes ranging from $N = 1{,}000$ to $N = 16{,}000$. Again, we tested our pre-trained RowDualNet model (trained only on synthetic data) directly on these matrices *without fine-tuning*.

**Analysis.** As shown in Fig. 5, our method generalizes remarkably well to the MOT distribution. We observe a consistent speedup $\approx 1.25\times$ relative to the faster of the baselines (LAP) for $N \geq 8{,}000$. We note that this speedup is naturally lower than that achieved on synthetic data: since MOT involves tracking objects between adjacent frames, the resulting matrices are relatively sparse. This sparsity presents a more difficult setting for our dense neural predictor compared to classical heuristics, yet our method still yields a distinct performance improvement. Notably, for SciPy's solver, a major speedup of $\approx 2\times$ is obtained. This result is significant, because real-world tracking matrices often contain complex correlations (e.g., spatial locality) that differ from our synthetic training data. The fact that RowDualNet accelerates these instances zero-shot confirms that it learns fundamental properties of the assignment structure rather than overfitting to synthetic distributions.

### 5.3. Runtime Analysis

**Runtime Breakdown and Overhead Analysis.** To verify that the neural warm-start does not dominate runtime, we performed a detailed breakdown of the end-to-end execution time. Table 1 reports the wall-clock time spent in each stage of the pipeline for the Dense Model. The results for Block-Structured Model (in App. D) exhibits nearly identical characteristics. The pipeline consists of five stages:

1. **Data Transfer:** Moving the cost matrix from CPU to GPU and back.

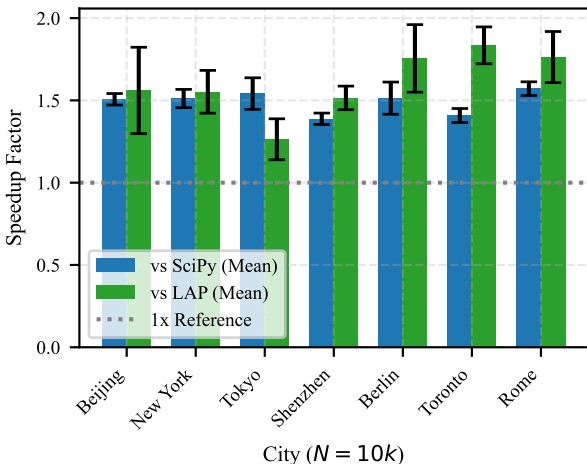

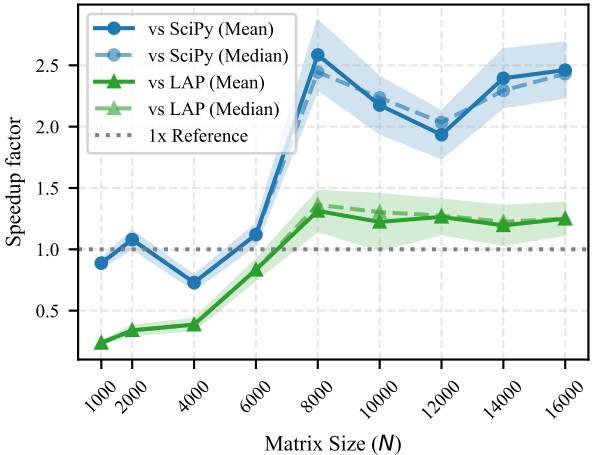

*Figure 4.* **Generalization to Urban Transportation Networks.** End-to-end speedup factor on cost matrices derived from Open-StreetMap (OSM) for $N = 10,000$ locations in major global metropolises. Our method achieves consistent acceleration ($\approx 1.2\times - 1.8\times$) across diverse urban topologies, demonstrating robustness to real-world road network structures without fine-tuning. The error bars indicate the 95% CI over 10 random instances.

*Figure 5.* **Generalization to Real-World Multiple Object Tracking.** End-to-end speedup factor on cost matrices extracted from the Large-Scale MOT dataset. The solid lines indicate the mean speedup, while the shaded regions represent the 95% CI. Even without fine-tuning, our model generalizes to the structured costs of vision-based tracking, achieving a consistent speedup of $\approx 1.25\times$ for $N \geq 8,000$ over the fastest baseline (LAP).

2. **Row Features:** Extracting the row-centric features.
3. **RowDualNet:** The forward pass of RowDualNet.
4. **Min-Trick:** Calculating $\hat{v}$ via the output of RowDualNet ($\hat{u}$).
5. **Seeded LAP:** Exact solver seeded with our duals.

**Analysis.** As shown in Table 1, the relative overhead of the neural components decreases as problem size increases. For massive dense instances ($N = 16,384$), the neural inference (Features + Model + Data transfer) consumes less than 7% of the total runtime, while the exact solver accounts for 93%. Similarly, for Block-Structured matrices (see Table 3 in App. D), the neural overhead is even lower, constituting less than 5% of the total runtime. Importantly, even when accounting for GPU acceleration, the dominant cost remains the CPU-based exact solver, indicating that the observed speedups stem from algorithmic improvements (fewer augmentations and dual updates) rather than from simple offloading to specialized hardware.

### 5.4. Runtime Stability

A key finding is that classical baseline runtime shows extreme variance across random seeds and matrix permutation (see App. I). This volatility stems from the sensitivity of augmenting-path algorithms to adversarial cost configurations, "unlucky" memory layouts can trigger worst-case pathfinding behavior, causing runtime spikes of up to $11\times$ (ratio of worst-case to best-case runtime). In contrast, our Neural Warm-Start consistently initializes the solver in a high-quality state, effectively stabilizing the optimization process. Neural Dual Warm-Starting not only improves mean runtime but drastically reduces the coefficient of variation (from $\approx 45\%$ to $\approx 30\%$), offering the predictable latency profile required for safety-critical real-time systems.

### 5.5. Mechanism and Ablation Analysis

To understand the source of the observed speedups and validate the necessity of our deep architecture, we conducted an extensive internal analysis (detailed in App. E and App. G).

*Table 1.* Average Runtime breakdown (ms) and percentage of total time for different matrix sizes $N \times N$ in Dense Model matrices.

| Component | $N = 512$ | | $N = 1024$ | | $N = 1536$ | | $N = 2048$ | | $N = 4096$ | | $N = 8192$ | | $N = 16384$ | |
|---|---|---|---|---|---|---|---|---|---|---|---|---|---|---|
| | Time | % | Time | % | Time | % | Time | % | Time | % | Time | % | Time | % |
| Data Transfer | 0.25 | 5.2 | 0.93 | 5.1 | 2.21 | 5.5 | 4.39 | 5.6 | 31.9 | 7.8 | 121 | 5.9 | 516 | 4.8 |
| Row Features | 0.62 | 12.8 | 2.21 | 12.1 | 1.67 | 4.1 | 2.39 | 3.1 | 7.52 | 1.8 | 66.8 | 3.3 | 181 | 1.7 |
| RowDualNet | 0.65 | 13.5 | 0.95 | 5.2 | 1.60 | 4.0 | 1.61 | 2.1 | 3.32 | 0.8 | 8.10 | 0.4 | 22.6 | 0.2 |
| Min-Trick | 0.09 | 1.8 | 0.09 | 0.5 | 0.11 | 0.3 | 0.15 | 0.2 | 0.34 | 0.1 | 1.07 | 0.1 | 3.87 | 0.0 |
| Seeded LAP | 3.21 | 66.6 | 14.1 | 77.0 | 34.7 | 86.1 | 69.8 | 89.1 | 368 | 89.5 | 1845 | 90.4 | 10138 | 93.3 |
| Total (ms) | 4.82 | 100.0 | 18.2 | 100.0 | 40.3 | 100.0 | 78.3 | 100.0 | 411 | 100.0 | 2042 | 100.0 | 10861 | 100.0 |

**Work Reduction.** We found that the primary driver of acceleration is the quality of the initial equality subgraph. While standard cold-start heuristics resolve only $\approx 26\%$ of assignments in the greedy phase, our neural warm-start allows the solver to match $\approx 76\%$ of assignments immediately (App. E), drastically pruning the search space, reducing the number of computationally expensive augmenting path searches by $\approx 68\%$ across all problem sizes.

**Necessity of Deep Learning (Ablation Study).** In App. G, we compared RowDualNet against linear regression and statistical heuristics (Random, Row Mean). Simple heuristics consistently failed to outperform the cold-start baseline (speedup $<1.0$). Furthermore, while linear models provided marginal gains at small scales, they failed to generalize to large instances, degrading below baseline performance for $N > 4,096$. Confirming that the non-linear feature extraction of RowDualNet is essential for capturing the complex dynamics of large-scale assignment problems.

## 5.6. Robustness Analysis

**Implementation Details.** We set the numerical tolerance $\epsilon = 10^{-5}$ to account for 32-bit floating-point precision. The density threshold was set to $\tau = 1.2$ (implying an expectation of at least 1.2 valid edges per row), calibrated to optimally balance safety and performance on a held-out validation set.

**Empirical Safety and Sensitivity.** During standard evaluation across the Dense, Block-Structured, and Real-World distributions, the fallback mechanism was rarely triggered. To ensure this low trigger rate is a result of high-quality dual predictions rather than an overly conservative threshold, we conducted a sensitivity analysis (detailed in App. J). By artificially perturbing optimal duals with Gaussian noise, we confirmed a direct correlation: as prediction quality drops, the equality subgraph density $\rho$ falls below $\tau$ precisely as solver runtime begins to degrade. This confirms the density criterion is a highly responsive quality metric.

## 6. Comparison with Dinitz et al. (2021) (Learned duals)

To empirically validate our architectural choices against the "Learned Duals" framework (Dinitz et al., 2021) (see Section 7.2 for an extended literature discussion), we implemented their "Learned Median" baseline. This method initializes duals using the coordinate-wise median of optimal duals observed in the training set.

Fig. 6 and Fig. 7 compare the speedup of our Neural Warm-Start against this baseline on both Block-Structured Model and Dense Model.

**Analysis.** On the **Block-Structured Model** dataset (Fig. 6),

the static "Learned Median" baseline from (Dinitz et al., 2021) fails to learn, dropping below $1.0\times$ in all instances. On the **Dense Model** dataset (Fig. 7), the Dinitz baseline provides negligible gain for larger instances, while our method maintains a robust $\approx 2.0\times$ speedup. The static median cannot account for the specific random noise realizations of each instance. It biases the solver towards a "mean" configuration that may be suboptimal for the specific graph. In contrast, RowDualNet adapts to the specific features of each instance, yielding superior performance in both cases.

## 7. Related Work

Our work sits at the intersection of deep learning and combinatorial optimization. Recent works have demonstrated the versatility of learning-augmented methods in combinatorial optimization settings (Mitzenmacher & Vassilvitskii, 2022; Khodak et al., 2022; Dinitz et al., 2022; 2024). Further details about these publications can be found in App. M as well as a discussion about classical baselines. Specifically, for the LAP framework, we categorize approaches into Neural Solvers (primal) and Warm-Start methods (dual).

### 7.1. Neural Combinatorial Optimization (Primal)

A growing body of work attempts to replace classical solvers with deep learning models, by formulating the LAP as a link prediction task. Liu et al. (2024) propose a graph network that predicts the permutation directly. Aironi et al. (2024) utilize Message-Passing GNNs for edge classification, while Loveland et al. (2025) employ line-graph transformations for multi-task assignment. Pan et al. (2024) introduced a self-supervised approach for bipartite matching.

While fast, these "neural solvers" suffer from two critical limitations. First, they are *approximations*, they treat hard constraints as soft loss terms or rely on greedy post-processing, failing to guarantee exact optimality. Second, they struggle with *scalability*. Architectures relying on edge convolutions (e.g., MAGNET (Loveland et al., 2025) or HybridGNN (Pan et al., 2024)) require $\mathcal{O}(N^2)$ memory, causing exhaustion on instances larger than $N \approx 2,000$. In contrast, our Neural Dual Warm-Starting architecture scales efficiently to large dense instances of $N = 16,384$.

### 7.2. Learning to Warm-Start (Dual)

Closer to our method is the approach of using machine learning to guide exact algorithms. The foundational work of Dinitz et al. (2021) introduced "learned duals" to warm-start the Hungarian algorithm.

**Theoretical Extensions.** Subsequently, Sakaue & Oki (2022) utilized Discrete Convex Analysis (DCA) to establish tighter bounds based on the predicted duals, while Chen et al. (2022) proposed a refined algorithm with error-dependent

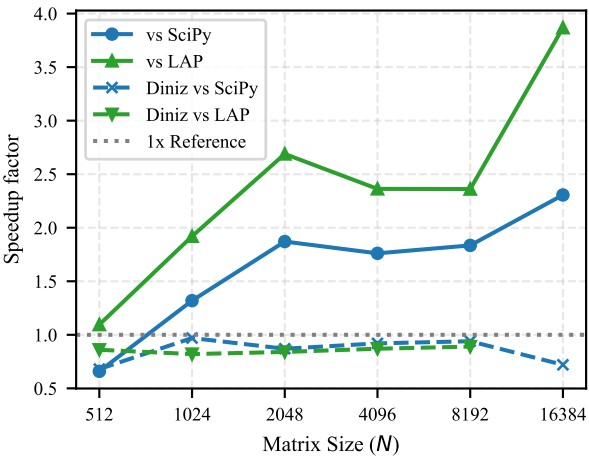

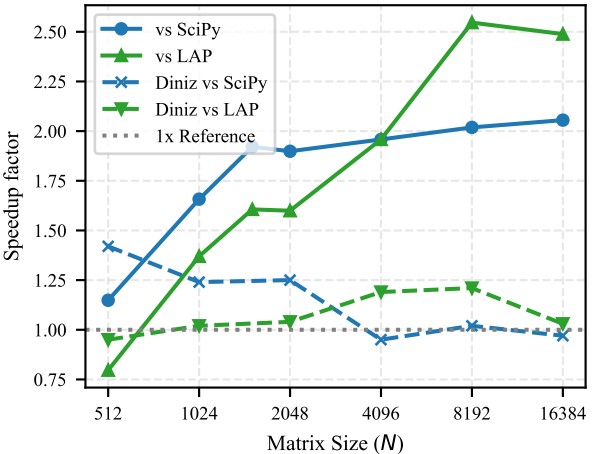

*Figure 6.* **Comparison with Dinitz et al. (2021) on Block-Structured Model.** Speedup factor relative to Cold-Start baselines (average of each problem size). On the Block-Structured Model dataset, the static "Learned Median" baseline from (Dinitz et al., 2021) (dashed lines) fails to scale, dropping below $1.0\times$ for all instances. In contrast, Neural Dual Warm-Starting (solid lines) achieves superior acceleration, reaching up to $\approx 3.5\times$ against LAP.

*Figure 7.* **Comparison with Dinitz et al. (2021) on Dense Model.** Speedup factor relative to Cold-Start baselines (average of each problem size). On the Dense Model dataset, the Dinitz baseline provides negligible gain, while our method maintains a robust speedup of $\approx 2.0\times - 2.5\times$.

complexity scaling with $\mathcal{O}(\sqrt{N})$. However, these works primarily focus on the *solver side*, proving worst-case guarantees assuming the existence of a predictor, and evaluate on small-scale instances ($N \approx 1{,}000$).

Our work complements this theoretical progress by addressing the *learning side*. We identify three practical limitations in prior frameworks:

1. **Feasibility Overhead:** Prior methods rely on iterative algorithms to repair infeasible predictions. We bypass this via the constructive Min-Trick.
2. **Expressivity:** Statistical heuristics (e.g., medians) relied upon by Dinitz et al. (2021) fail on unstructured or row-permuted data (as shown in Sec. 6). RowDualNet captures complex instance-specific dependencies.
3. **Scale:** We demonstrate robustness on matrices up to $N = 16{,}384$, substantially larger than previous studies.

## 8. Conclusion

We presented **Neural Dual Warm-Starting**, a learning-augmented framework that accelerates exact LAP solvers by predicting continuous dual potentials. Unlike primal-based neural solvers, our approach guarantees feasibility and scales to $N = 16{,}384$ without exhausting GPU memory. Empirically, our method yields end-to-end speedups of up to $4\times$ on structured synthetic instances and demonstrates robust zero-shot generalization to real-world transportation and tracking tasks ($\approx \mathbf{1.25\times - 1.8\times}$). Crucially, it stabilizes runtime performance, drastically reducing the extreme

variance observed in classical solvers.

Future work includes extending this "dual warm-start" paradigm to other combinatorial problems such as Min-Cost Flow and exploring semi-supervised training regimes where optimal duals are not available. By bridging the gap between the speed of deep learning and the guarantees of exact optimization, our framework paves the way for real-time decision-making in large-scale industrial systems.

## Acknowledgements

The research of I. Yavlovich and N. Weinberger was supported by the United States – Israel Binational Science Foundation (NSF-BSF), grant no. 2024763.

## Impact Statement

This paper presents work whose goal is to advance the field of Machine Learning and Combinatorial Optimization. Our primary contribution is a framework for accelerating exact solvers for the Linear Assignment Problem (LAP), a fundamental algorithmic primitive.

There are many potential societal consequences of our work, both positive and negative. On the positive side, the LAP is central to logistics, supply chain management, and resource allocation. Accelerating these computations at scale can lead to significant efficiency gains in transportation networks, potentially reducing fuel consumption and carbon footprints. Furthermore, faster assignment algorithms facilitate more responsive real-time decision-making in emergency response and disaster relief logistics.

On the negative side, the LAP is a core component of Multi-Object Tracking (MOT) systems used in computer vision. Improvements in the speed and scalability of exact matching could inherently enhance the capabilities of automated surveillance and personnel tracking technologies. While our work focuses on the algorithmic and mathematical foundations of optimization, we acknowledge that the downstream deployment of such efficient tracking systems warrants careful ethical consideration and regulation to protect privacy.

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

## Appendix Outline

This appendix provides supplementary theoretical analysis, implementation details, and extended experimental results to support the main paper. We begin by formally analyzing the asymptotic time complexity of our pipeline in App. A, followed by detailed definitions of our input features and training hyperparameters in App. B and App. C.

To validate the source of our performance gains, we present the detailed runtime breakdown for block-structured matrices in App. D, investigate the internal reduction of the solver's search space in App. E, and analyze the effect of matrix sparsity on overall acceleration in App. F. We further justify our architectural choices through extensive ablation studies in App. G, which evaluate alternative warm-start strategies and classical subgradient initialization, followed by an analysis of feature dimension selection in App. H. Finally, we comprehensively evaluate the system's reliability and safety through a runtime variance analysis in App. I, a sensitivity of the density test in App. J, hyperparameter stability checks in App. K, and a verification of permutation invariance in App. L, concluding with an extended discussion of related work in App. M.

## A. Detailed Time Complexity Analysis

We analyze the computational cost of our pipeline to demonstrate that the neural overhead is negligible for large-scale instances. The standard cold-start LAPJV algorithm has a worst-case time complexity of $\mathcal{O}(N^3)$.

The overhead introduced by our Neural Warm-Start consists of three components:

1. **Feature Extraction:** Computing row statistics requires scanning the full cost matrix once, an $\mathcal{O}(N^2 \log(N))$ operation (worst-case due to sorting each row).
2. **RowDualNet Inference:** Our architecture processes $N$ rows in parallel using a shared MLP with fixed hidden dimension $H$. The complexity is $\mathcal{O}(N \cdot H)$, which is linear with respect to the number of agents.
3. **Min-Trick:** Computing $\hat{v}$ requires finding the minimum over each column, which involves scanning the matrix, an $\mathcal{O}(N^2)$ operation.
4. **Fallback Mechanism:** Computing the assignment average degree $\rho$ requires evaluating the reduced cost condition for edges in the matrix, an $\mathcal{O}(N^2)$ operation.

The total worst-case overhead is $T_{overhead} = \mathcal{O}(N^2 \log(N))$. Since the solver complexity is $\mathcal{O}(N^3)$, the overhead is asymptotically dominated by the solver's $\mathcal{O}(N^3)$ runtime as $N$ increases. In practice, the warm-start reduces the constant factor of the solver's $\mathcal{O}(N^3)$ runtime by initializing the search closer to the optimum, resulting in the net speedups observed in our experiments in Sec. 5.

## B. Detailed Feature Definitions

Our **Neural Dual Warm-Starting** architecture relies on a compact set of 21 row-centric features, listed in Table 2, to predict the dual potentials $\hat{u}_i$. These features are designed to be invariant to column permutations, ensuring that the model generalizes across different task orderings. We categorize them into four groups: Distributional Statistics, Ambiguity Metrics, Competitiveness Signals, and Positional Encodings.

**Remark on Permutation Invariance.** While our row-centric features are designed to be column-permutation invariant, the inclusion of sinusoidal Positional Encodings (PE) technically breaks row-permutation invariance. We include PEs to break symmetry in degenerate cases (e.g., identical cost rows in synthetic data), where a strictly invariant network would be forced to predict identical potentials, potentially hindering the solver. In practice, we observe that the network learns to rely primarily on cost statistics, using PEs only as a tie-breaking signal.

*Table 2.* Details of the 21 input features used in the Neural Dual Warm-Starts.

| Category | Dim | Feature Name | Description |
|---|---|---|---|
| **Distribution** | 4 | `row_min` | The minimum cost in the row ($C_{i,\min}$). |
| | | `row_max` | The maximum cost in the row. Provides the scale/range of costs. |
| | | `row_mean` | The arithmetic mean of all costs for agent $i$. |
| | | `row_std` | The standard deviation of costs, indicating cost variability. |
| **Ambiguity** | 2 | `entropy` | Shannon entropy of the softmax-normalized row costs. A high entropy indicates an agent is "indifferent" between many tasks while low entropy indicates a strong preference for specific tasks. |
| | | `difficulty` | Inverse of the mean gap between sorted costs. Captures how "clumped" the costs are (harder to distinguish). |
| **Competition** | 2 | `near_best` | The fraction of tasks with costs within 10% of the row minimum. Indicates flexibility (how many "good" options exist). |
| | | `is_col_best` | The normalized count of columns $j$ for which this row $i$ holds the global minimum cost ($C_{ij} = \min_k C_{kj}$). For capturing global competition without explicit message passing. |
| **Local Context** | 5 | `k_mean` | Mean of the $K = 10$ smallest costs. Focuses the network on the "relevant" cheapest tasks rather than the global average. |
| | | `k_std` | Standard deviation of the $K = 10$ smallest costs. |
| | | `rank_mean` | The average rank of this agent across all columns. |
| | | `rank_std` | Standard deviation of the agent's ranks. |
| | | `norm_rank` | Normalized rank score, providing a smoothed competitiveness metric. |
| **Positional** | 8 | `sin/cos` encodings | Sinusoidal position embeddings for row indices (freqs 1, 2, 4, 8). Allows the network to distinguish between agents that might have identical cost statistics. |
| **Total** | 21 | | |

# C. Detailed Architecture and Training

We train RowDualNet (illustrated in Fig. 8) using a supervised approach with ground-truth duals generated by the LAPJV solver.

- **Architecture:** A residual MLP with 3 blocks. Hidden dimension $H = 192$.
- **Optimizer:** AdamW with weight decay $1 \times 10^{-4}$.
- **Learning Rate:** Initial LR $1 \times 10^{-3}$ with a ReduceLROnPlateau scheduler (factor 0.5, patience 10).
- **Loss Function:** $\mathcal{L} = \text{MAE}(\hat{u}, u^*) + \lambda \sum_{(i,j) \in M^*} \text{ReLU}(C_{ij} - \hat{u}_i - \hat{v}_j)$, where the second term serves as a Complementary Slackness regularizer to enforce structural validity by minimizing the violation of primal-dual tightness on ground-truth optimal edges.
- **Global Context:** We use Top-K pooling with $K = 16$ during the `sparse_refine` step to inject column-side competition info.

**Optimization Note.** The Min-Trick operation $v_j = \min_i(C_{ij} - u_i)$ is continuous and differentiable almost everywhere. During backpropagation, gradients flow through the $\arg\min$ index $i^*$ for each column ($v_j \rightarrow u_{i^*}$). While non-differentiable

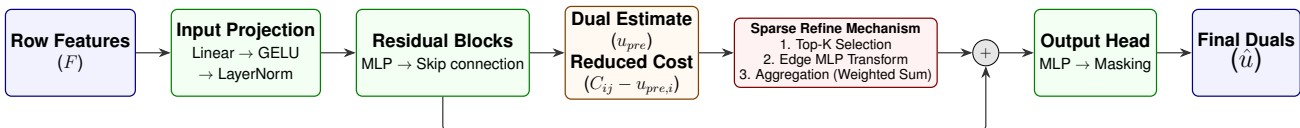

*Figure 8.* **RowDualNet Architecture.** The model processes row features through a residual MLP to produce an initial dual estimate $\hat{u}_{pre}$. The Sparse Refine Mechanism (center) then injects global context by inspecting the top-$K$ most competitive columns (lowest reduced cost) for each agent, allowing the network to adjust bids based on contention before the final output.

at exact ties, standard subgradient handling in automatic differentiation frameworks (PyTorch) proved sufficient for stable convergence without explicit smoothing.

**Gauge-Fixing Dual Labels.** To address the inherent translation invariance of dual variables (where shifting all $u$ and $v$ by a constant yields the same reduced costs), we apply gauge-fixing to the dual labels during training. By centering the optimal row potentials, we ensure a stable, unique learning target for the network.

## D. Runtime Breakdown for Block-Structured Model

Table 3 provides the detailed runtime breakdown for the Block-Structured Model. Consistent with the Dense Model results presented in Sec. 5.3, the neural overhead remains negligible for large-scale instances. Specifically, at $N = 16{,}384$, the neural components (data transfer, row feature extraction, and RowDualNet inference) account for less than 5% of the total wall-clock time, verifying that the speedup is driven by the efficient initialization of the solver rather than hardware acceleration.

*Table 3.* Average Runtime breakdown (ms) and percentage of total time for different matrix sizes $N \times N$ in Block-Structured Model matrices.

| Component | $N = 512$ | | $N = 1024$ | | $N = 2048$ | | $N = 4096$ | | $N = 8192$ | | $N = 16384$ | |
|---|---|---|---|---|---|---|---|---|---|---|---|---|
| | Time | % | Time | % | Time | % | Time | % | Time | % | Time | % |
| Data Transfer | 0.66 | 2.6 | 5.35 | 7.3 | 10.9 | 4.6 | 71.0 | 5.7 | 237 | 4.0 | 863 | 3.8 |
| Row Features | 3.30 | 13.2 | 2.36 | 3.2 | 3.10 | 1.3 | 10.4 | 0.8 | 54.7 | 0.9 | 210 | 0.9 |
| Tensor Prep. | 0.01 | 0.0 | 0.01 | 0.0 | 0.01 | 0.0 | 0.01 | 0.0 | 0.01 | 0.0 | 0.01 | 0.0 |
| RowDualNet | 1.80 | 7.2 | 1.98 | 2.7 | 2.07 | 0.9 | 4.13 | 0.3 | 9.45 | 0.2 | 26.6 | 0.1 |
| Min-Trick | 0.19 | 0.8 | 0.16 | 0.2 | 0.22 | 0.1 | 0.48 | 0.0 | 1.27 | 0.0 | 4.55 | 0.0 |
| Seeded LAP | 19.1 | 76.2 | 63.6 | 86.6 | 223 | 93.2 | 1159 | 93.1 | 5639 | 94.9 | 21756 | 95.2 |
| Total (ms) | 25.1 | 100.0 | 73.5 | 100.0 | 240 | 100.0 | 1245 | 100.0 | 5942 | 100.0 | 22860 | 100.0 |

## E. Work Reduction

To understand the source of the observed speedups, we inspected the internal state of the LAPJV solver. Specifically, we measured the Greedy Match Rate, i.e. the percentage of rows that can be validly matched immediately using the initial duals, without requiring any augmenting path search.

We analyzed this metric on the structured Block-Structured Model dataset and Dense Model across all problem sizes and compared it to `SciPy`. Fig. 9 and Fig. 10 present the results.

**Analysis.** The standard Cold–Start heuristics successfully resolve only $\approx 26\%$ of the assignments in the first phase (for both models), leaving the majority of the problem ($\approx 74\%$) to be solved by the expensive shortest-augmenting-path phase. In contrast, our RowDualNet predictions provide high-quality duals that $\approx 76\%$ of the assignments are resolved instantly in the equality subgraph (for both models). The solver then only needs to run the expensive augmentation logic for the remaining 24% of rows. This $\approx 68\%$ reduction in the search space (Fig. 9(right) and Fig. 10(right)) is the primary driver of the $2.0\times - 2.5\times$ speedup observed in Sec. 5.2.

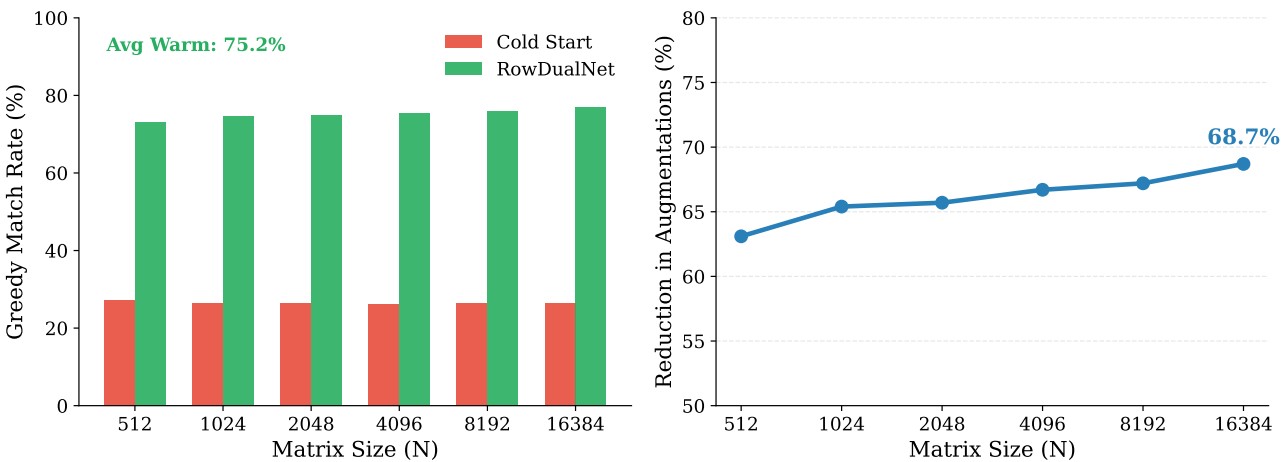

*Figure 9.* **Solver Work Analysis for Block-Structured Model.** (Left) The average percentage of rows matched during the initial greedy phase. Standard Cold-Start heuristics (`SciPy`, Red) consistently resolve only $\approx 26\%$ of assignments. In contrast, our Neural Warm-Start (Green) matches 75.2% of agents immediately. (Right) This superior initialization drastically reduces the number of "free rows" requiring expensive augmenting path searches, cutting the solver's combinatorial search effort by 68.7%.

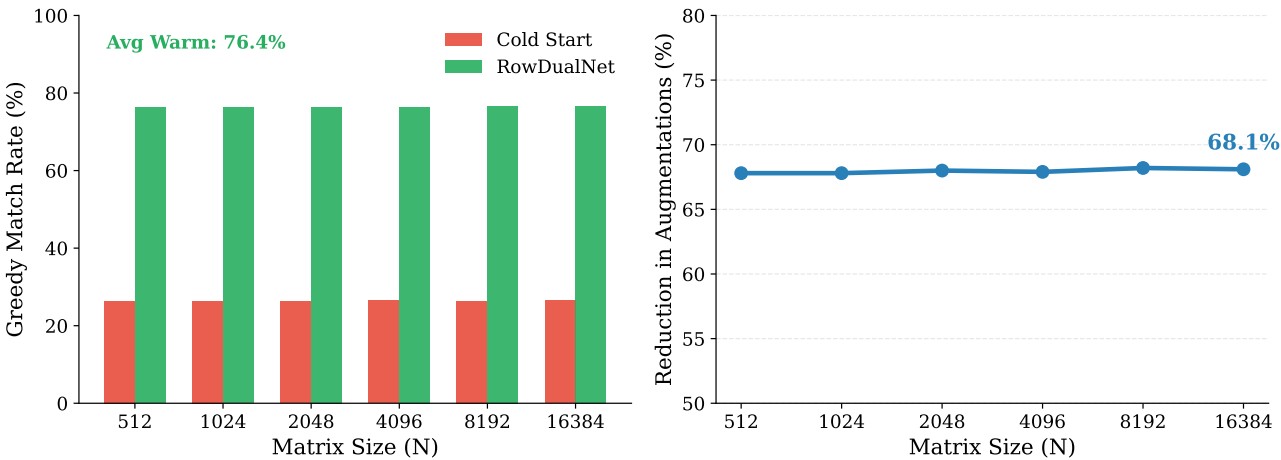

*Figure 10.* **Solver Work Analysis for Dense Model.** (Left) The average percentage of rows matched during the initial greedy phase. Standard Cold-Start heuristics (`SciPy`, Red) consistently resolve only $\approx 27\%$ of assignments. In contrast, our Neural Warm-Start (Green) matches 76.4% of agents immediately. (Right) This superior initialization drastically reduces the number of "free rows" requiring expensive augmenting path searches, cutting the solver's combinatorial search effort by 68.1%.

## F. Effect of Matrix Sparsity on Acceleration

To further understand the operating regime where our method provides the most benefit, we evaluated how the end-to-end acceleration of Neural Dual Warm-Starts correlates with matrix density.

**Experimental Setup.** We generated dense matrices of size $N = 4,096$ and $N = 8,192$ and progressively increased their sparsity by masking a percentage of the edges (ranging from $10\%$ up to $90\%$ sparsity). We then measured the end-to-end speedup factor of our method relative to both the SciPy and LAP cold-start baselines.

**Analysis.** As illustrated in Figure 11, the relative speedup factor exhibits a clear inverse correlation with matrix sparsity. On highly dense matrices ($10\%$ sparsity), our method achieves maximum acceleration ($\approx 2.5\times$). As the sparsity progressively increases toward $90\%$, the speedup margin narrows, gradually converging toward the $1.0\times$ cold-start baseline.

This behavior is expected: for highly sparse matrices, the total execution time of the classical LAPJV solver is already minimal because the limited number of valid edges naturally restricts the combinatorial search space. Consequently, the

margin for relative speedup is inherently limited. Conversely, on dense matrices where the search space is vast and "price wars" are frequent, the neural warm-start yields its maximum benefit.

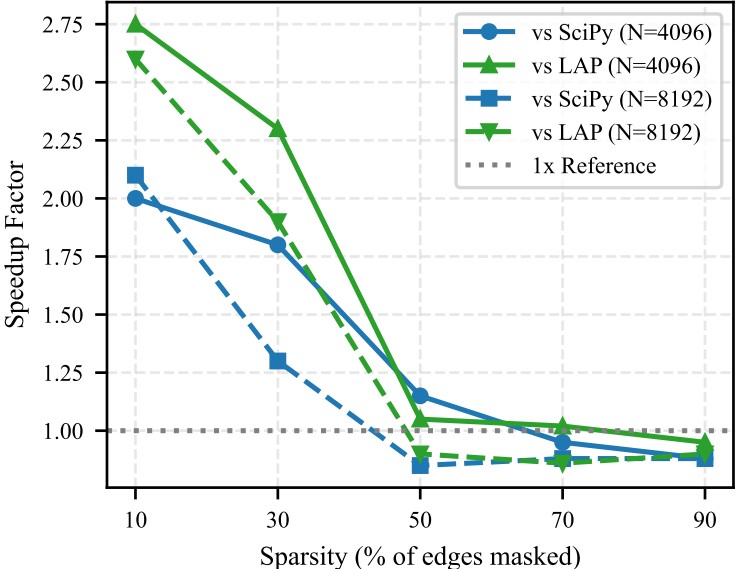

*Figure 11.* Effect of matrix sparsity on the end-to-end speedup factor for $N = 4,096$ and $N = 8,192$. As the percentage of masked edges increases, the absolute runtime of the classical solver drops, naturally limiting the margin for relative speedup.

## G. Ablation Study: Is Deep Learning Necessary?

A critical question is whether a deep neural network is truly required, or if simpler statistical models could achieve similar warm-start benefits. We compared our RowDualNet architecture against three alternative warm-start strategies while keeping the rest of the pipeline (Min-Trick + LAPJV) identical:

1. **Row Mean Heuristic:** Initializing $u_i = \text{mean}(C_{i,:})$, thus testing if simple distributional statistics are sufficient.
2. **Random Heuristic:** Initializing $u_i \sim U(0, 1)$, thus serving as a sanity check to measure the sensitivity of the solver to noise.
3. **Linear Regression:** Replacing the deep RowDualNet encoder with a single linear layer $\hat{u} = Wx + b$, thus acting as a proxy for "shallow" learning approaches with lack of non-linear expressivity.

*Note: We omit a 'Row Minimum' baseline ($u_i = \min_j C_{ij}$) because this corresponds exactly to the standard initialization phase of the LAPJV algorithm itself. Thus, the 'Row Min' performance is identical to the Cold-Start baseline ($1.0\times$).*

**Results.** As shown in Fig. 12 and Fig. 13, simple heuristics and linear models fail to generalize to large-scale instances. First, the Linear Regression model (pink) provides marginal speedups at small scales but degrades as $N$ grows, eventually becoming slower than the best cold start run at $N \geq$ 4,096. This indicates that the mapping from local cost statistics to optimal global duals involves complex non-linear dependencies that a simple linear projection cannot capture. Second, the Heuristic (Random) baseline (yellow) consistently performs worse than the cold start (Speedup $<1.0$). Confirming that the LAPJV solver is sensitive to initialization, a "bad" seed effectively worsens the search, forcing the solver to perform extra work to correct the duals. In contrast, Neural Dual Warm-Starts (orange) is the only method that maintains robust positive speedups across all scales, justifying the computational overhead of our deep architecture.

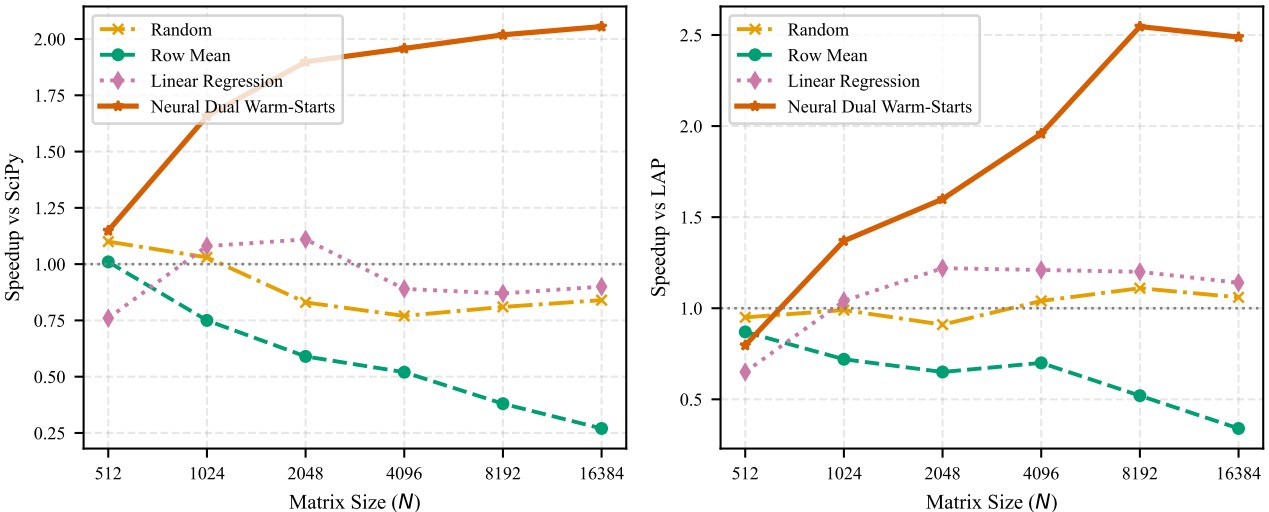

*Figure 12.* **Ablation Study on Dense Model.** Comparison of warm-start strategies (Mean of Ratios) relative to the Cold-Start baseline ($1.0\times$). While Linear Regression (pink) offers marginal gains for small instances, its performance degrades as $N$ increases, confirming that simple models cannot capture the complexity of large, dense assignment problems. Heuristics like Random and Row Mean (yellow/green) consistently perform worse than the cold start ($<1.0\times$), actively hindering the solver. Only Neural Dual Warm-Starts (orange) maintains robust acceleration, reaching $\approx 2.0\times$ speedup at scale.

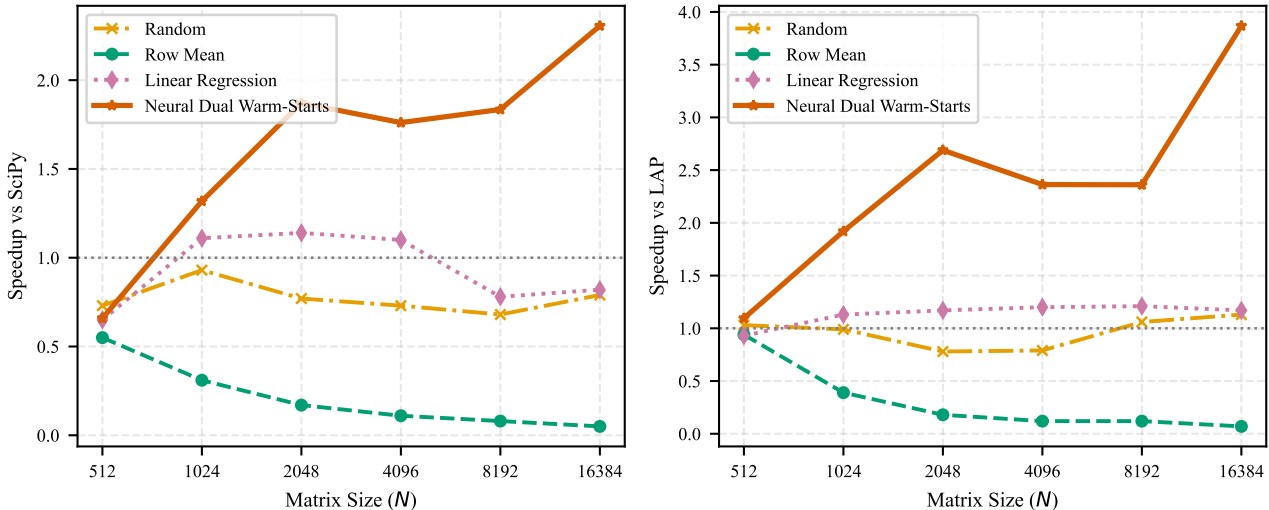

*Figure 13.* **Ablation Study on Block-Structured Model.** Speedup factors on Block-Structured Model instances (Mean of Ratios). The Linear Regression baseline (pink) fails to generalize as problem size grows, eventually falling below the cold-start baseline ($1.0\times$) at large $N$. In contrast, Neural Dual Warm-Starts (orange) successfully exploits the latent group structure, achieving superior speedups of up to $\approx 3.5\times$ against `LAP`. This confirms that deep non-linear feature extraction is essential for disentangling complex cost topologies.

### G.1. Comparison with Classical Subgradient Initialization

While simple statistical heuristics fail to scale, we tested whether a classic iterative optimization technique, such as subgradient ascent on the dual objective, could provide a comparable warm-start without requiring deep learning.

**Experimental Setup.** We implemented a subgradient ascent dual initialization baseline. To ensure a strictly fair comparison, we bounded the execution time of the subgradient method to exactly match the forward-pass inference time of RowDualNet. We evaluated this classical warm-start against our Neural Warm-Start across all four dataset topologies: Dense ($N = 8,192$), Block-Structured ($N = 8,192$), OpenStreetMap ($N = 10,000$), and MOT ($N > 6,000$).

**Analysis.** As shown in Figure 14, RowDualNet shows significant performance improvements over the classical subgradient approach across all dataset scenarios. On the dense and structured datasets (Dense and Block-Structured), our neural method achieves an average speedup of $\approx 2.5\times$ over a cold start, whereas the time-bounded subgradient method plateaus at $\approx 1.5\times$ speedup.

Furthermore, on the relatively sparse Multi-Object Tracking (MOT) dataset, the subgradient method actively degrades performance, yielding inferior results compared to the basic cold-start baseline (speedup $< 1.0\times$). Because sparse matrices execute very rapidly under classical heuristics, any inaccurate dual initialization overhead is heavily penalized. Our deep learning approach, however, successfully navigates this sparsity, reliably maintaining positive acceleration without falling into the subgradient degradation trap.

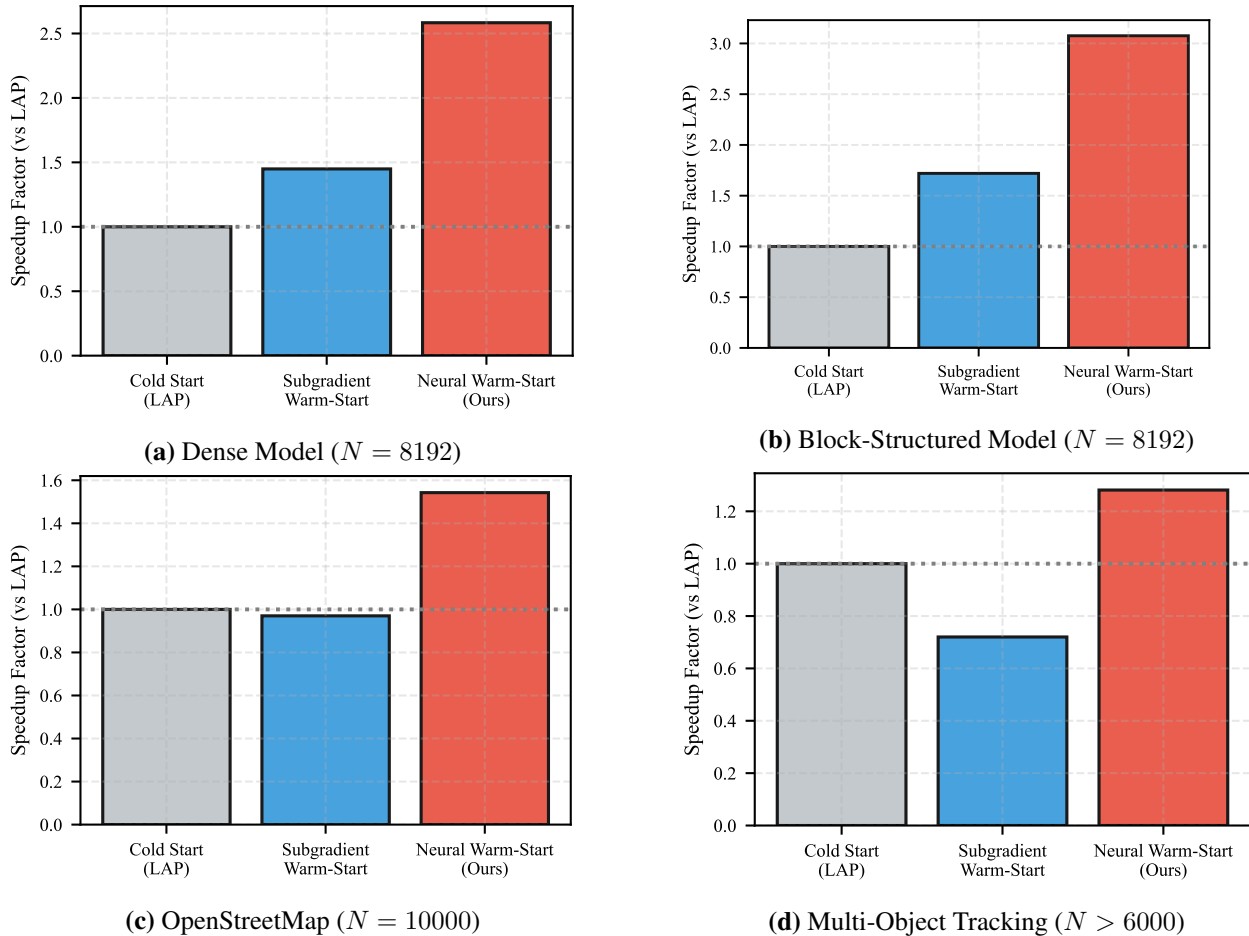

(a) Dense Model ($N = 8192$)

(b) Block-Structured Model ($N = 8192$)

(c) OpenStreetMap ($N = 10000$)

(d) Multi-Object Tracking ($N > 6000$)

*Figure 14.* Ablation comparing Neural Dual Warm-Starts against a time-bounded Subgradient Ascent initialization. **(a)** and **(b)** demonstrate that on dense and structured synthetic data, our method yields an average speedup of $\approx 2.5\times$, significantly outperforming the subgradient baseline. **(c)** shows sustained acceleration on urban topologies. Crucially, **(d)** illustrates that on highly sparse data (MOT), classical subgradient initialization degrades performance (speedup $< 1.0\times$), whereas RowDualNet maintains positive acceleration.

## H. Feature Dimension and Selection

A key advantage of our architecture is that it reduces the input dimension from $\mathcal{O}(N^2)$ to $\mathcal{O}(d \cdot N)$, avoiding the quadratic memory bottleneck of graph-based approaches. The exact number of features is not crucial, provided that $d$ is a relatively small constant that does not scale with the problem dimension $N$.

Naturally, there is a trade-off between the number of features and the coverage of the model. To evaluate this trade-off, we ablated our feature set and trained RowDualNet with varying capacities:

1. **Simple Stats** ($d = 4$)**:** Only basic distributional statistics (row min, max, mean, and std).

2. **No Positional Encoding** ($d = 13$)**:** All formulated features excluding the sinusoidal positional encodings.

3. **Full Features** ($d = 21$)**:** The complete feature set utilized in our main experiments.

**Analysis.** We evaluated the end-to-end speedup of these variants on both the Dense and Block-structured datasets at $N = 4,096$ (Figure 15). On the Dense model, both the $d = 13$ and $d = 21$ sets yield similar speedups, consistently outperforming the simple $d = 4$ baseline. However, on the more challenging Block-structured matrices, where finer discrimination between similar rows is required, the inclusion of positional encodings ($d = 21$) provides significant performance gains by effectively breaking symmetries. Overall, the choice of $d = 21$ represents an empirical sweet spot: it balances expressiveness and efficiency, enabling robust performance on difficult structured instances while keeping the feature dimension $d$ strictly small and constant.

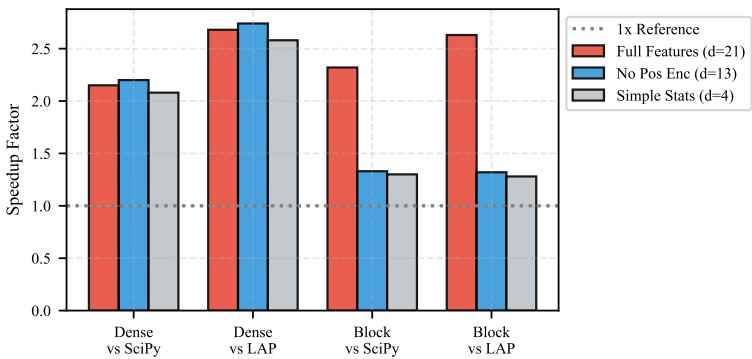

*Figure 15.* Feature ablation on Dense and Block-structured matrices ($N = 4,096$). While simple statistics ($d = 4$) provide minimal gains, the full feature set ($d = 21$) is required to achieve maximum speedup, particularly on highly structured distributions where positional encodings aid in symmetry breaking.

# I. Stability Analysis

To assess the suitability of assignment algorithms for latency-critical applications (such as real-time tracking), we evaluated the runtime stability of the two state-of-the-art baselines against our proposed method. We benchmarked all solvers on the MOT dataset (which contains complex correlations) with 30 trials per problem size.

**Baseline Instability.** As shown in Table 5, `SciPy` exhibits significant runtime instability, with a Coefficient of Variation (CV) exceeding 40% for large-scale problems ($N \geq 8,000$). The C++ `LAP` library (Table 6), while faster on average, demonstrates even more extreme volatility. At $N = 10,000$, the LAPJV runtime fluctuates between $0.89$s and $10.20$s, a factor of over $11\times$ (ratio of worst-case to best-case runtime), resulting in a massive CV of $95.5\%$. This extreme unpredictability arises because heuristic initializations are highly sensitive to the specific memory layout and configuration of the input cost matrix.

**Stability via Learning.** In stark contrast, our Neural Dual Warm-Starting solver (Table 4) maintains a consistently low CV ($\approx 27 - 35\%$) across all scales. At $N = 16,000$, the CV of our approach is $27.8\%$, compared to $41.3\%$ for `SciPy` and $49.2\%$ for `LAP`. This empirically proves that the neural network "stabilizes" the solver. This predictable latency profile makes our method uniquely suitable for real-time guarantees.

*Table 4.* Runtime Variance of Neural Dual Warm-Starting on MOT Data (30 Trials)

| N | Mean (s) | CV (%) | Min (s) | Max (s) |
|---|---|---|---|---|
| 500 | 0.0033 | 12.12% | 0.0031 | 0.0047 |
| 1000 | 0.0080 | 8.75% | 0.0072 | 0.0100 |
| 1500 | 0.0185 | 15.68% | 0.0151 | 0.0268 |
| 2000 | 0.0365 | 31.23% | 0.0270 | 0.0731 |
| 4000 | 0.2582 | 35.36% | 0.1513 | 0.4670 |
| 6000 | 0.5654 | 29.04% | 0.4560 | 1.8881 |
| 8000 | 0.9863 | 33.64% | 0.9207 | 3.1372 |
| 10000 | 2.1132 | 32.64% | 1.6962 | 5.4201 |
| 12000 | 3.8998 | **29.05%** | 2.7937 | 8.5711 |
| 14000 | 5.3839 | **31.68%** | 5.0887 | 13.2564 |
| 16000 | 8.0233 | **27.75%** | 6.7022 | 20.0959 |

*Table 5.* Runtime Variance of `SciPy` on MOT Data (30 Trials)

| N | Mean (s) | CV (%) | Min (s) | Max (s) |
|---|---|---|---|---|
| 500 | 0.0014 | 13.84% | 0.0011 | 0.0021 |
| 1000 | 0.0071 | 19.87% | 0.0057 | 0.0119 |
| 1500 | 0.0207 | 20.72% | 0.0161 | 0.0337 |
| 2000 | 0.0394 | 17.35% | 0.0303 | 0.0612 |
| 4000 | 0.1882 | 21.35% | 0.1343 | 0.2707 |
| 6000 | 0.6334 | 29.04% | 0.4264 | 1.0413 |
| 8000 | 2.2909 | **45.68%** | 0.9733 | 4.2924 |
| 10000 | 4.5970 | **47.02%** | 1.9871 | 9.5304 |
| 12000 | 7.5427 | 45.06% | 3.7714 | 14.9113 |
| 14000 | 12.1742 | 41.66% | 6.4478 | 22.7414 |
| 16000 | 18.9621 | 41.28% | **10.0780** | **37.6409** |

*Table 6.* Runtime Variance of `LAP` on MOT Data (30 Trials)

| N | Mean (s) | CV (%) | Min (s) | Max (s) |
|---|---|---|---|---|
| 500 | 0.0004 | 7.38% | 0.0004 | 0.0005 |
| 1000 | 0.0019 | 5.33% | 0.0018 | 0.0021 |
| 1500 | 0.0049 | 12.3% | 0.004 | 0.0063 |
| 2000 | 0.0124 | 50.82% | 0.0076 | 0.029 |
| 4000 | 0.0998 | 50.68% | 0.0398 | 0.2182 |
| 6000 | 0.4714 | 43.43% | 0.1925 | 0.9830 |
| 8000 | 1.1638 | 56.04% | 0.4957 | 3.1264 |
| 10000 | 2.5853 | **95.51%** | **0.8934** | **10.1964** |
| 12000 | 4.1512 | 53.27% | 1.5650 | 9.8447 |
| 14000 | 6.0758 | 63.76% | 2.2199 | 13.9578 |
| 16000 | 9.6251 | 49.16% | 5.5396 | 20.1037 |

## J. Sensitivity of the Density

To demonstrate that our chosen average degree metric ($\rho$) is a meaningful indicator of prediction quality rather than an arbitrary conservative threshold, we conducted a sensitivity analysis. We disabled the fallback mechanism and artificially perturbed the optimal dual predictions with increasing levels of Gaussian noise. As shown in Figure 16, there is a direct correlation: as the noise increases, the equality subgraph density ($\rho$) drops, and the solver runtime rises exponentially. This

confirms that the density criterion is a highly responsive indicator of prediction quality. The fact that the fallback was rarely triggered during our main benchmarks indicates the consistently high quality of the initial dual predictions, not a lack of sensitivity in the trigger.

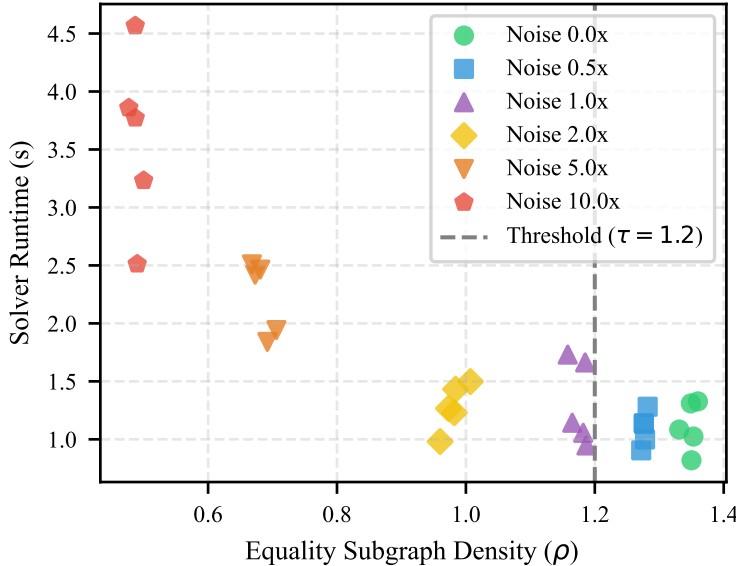

*Figure 16.* Sensitivity analysis showing that as artificial noise increases, the subgraph density drops and solver runtime rises, validating $\rho$ as a robust quality metric.

## K. Sensitivity to the Top-$K$ Hyperparameter

A core component of the RowDualNet architecture is the Sparse Refine Mechanism, which aggregates competitive information from the top-$K$ columns with the lowest reduced costs. To ensure our framework is robust to the choice of this hyperparameter, we evaluated the end-to-end speedup factor across a range of $K$ values.

**Analysis.** As illustrated in Figure 17, the empirical performance of RowDualNet remains remarkably stable when $K$ is kept as a relatively small constant (e.g., $K \in [8, 32]$). The network successfully extracts sufficient competitive signal from these top few edges to produce high-quality duals. By contrast, scaling $K \to N$ (i.e., aggregating over the entire row) actively diminishes the total end-to-end speedup. This degradation occurs because calculating the full rank introduces an $\mathcal{O}(N \log N)$ sorting overhead per row, which severely impacts the preprocessing time without providing any significant corresponding gain in dual prediction quality. Consequently, utilizing a small, constant $K$ is optimal for both performance and scalability.

## L. Permutation Invariance and Index Bias

As detailed in our feature definitions, RowDualNet utilizes sinusoidal positional encodings. The specific role of these features is to break symmetry in "extreme" or degenerate cases where multiple agents might share identical cost statistics, preventing the network from predicting identical potentials for distinct agents. However, it is crucial that these positional features do not introduce harmful spatial or ordering biases.

**Analysis.** To verify that the model maintains permutation invariance in practice, we conducted a sensitivity test. We selected fixed base cost matrices and evaluated each across 10 random row-wise permutations. As shown in Figure 18, the resulting solver running times across the permuted instances are highly stable, with a standard deviation of $\sigma \leq 0.24$ seconds in all tested cases for both $N = 4,096$ and $N = 8,192$. This tight variance confirms that the positional features successfully act as a symmetry-breaking mechanism without binding the network's predictions to the arbitrary original indexing of the input matrix.

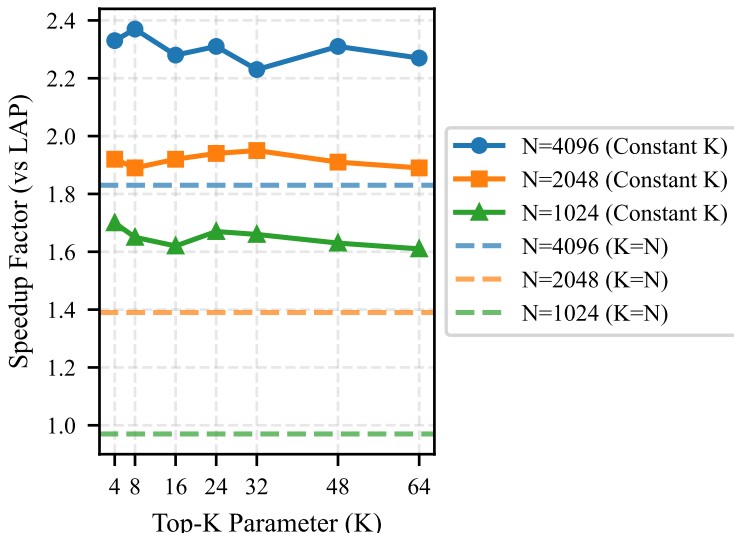

*Figure 17.* Sensitivity of the end-to-end speedup to the Top-$K$ hyperparameter. Performance remains highly stable for small constant values of $K$. Expanding the search to $K = N$ introduces unnecessary sorting overhead that degrades the overall system speedup.

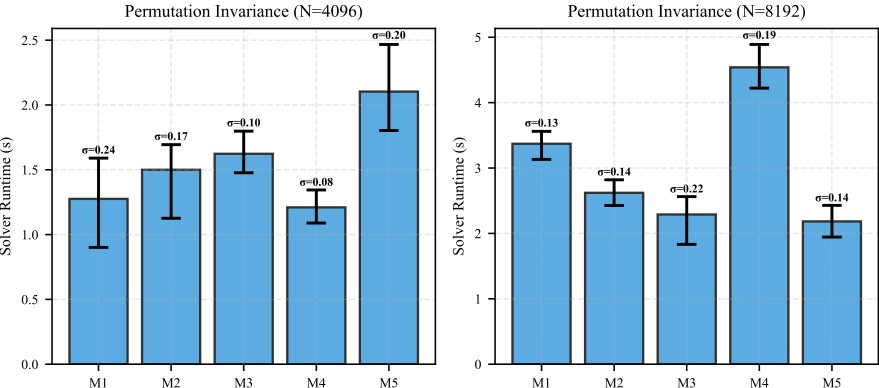

*Figure 18.* Evaluating permutation invariance. The end-to-end solver runtime remains highly stable ($\sigma \leq 0.24$) when the same cost matrix is subjected to random row-wise permutations, confirming that positional encodings do not introduce harmful index bias.

# M. Extended Related Work

This section provides additional context on related approaches that were summarized in Sec. 7.

## M.1. Classical and Algorithmic Baselines

The standard exact solvers for the LAP remain the Hungarian algorithm (Kuhn, 1955) and the Jonker-Volgenant (LAPJV) algorithm (Jonker & Volgenant, 1987). While LAPJV is significantly faster in practice than the Hungarian method, both suffer from $\mathcal{O}(N^3)$ worst-case complexity. Purely algorithmic works have sought to speed up assignment without learning. For example, Kriege et al. (2019) achieves linear-time assignment but only for costs forming a Tree Metric.

More relevant to dense instances are modern Auction algorithms, specifically, the aggressive $\epsilon$-scaling and cooperative bidding frameworks proposed by Bertsekas (2024). These methods are designed to mitigate "price wars" (oscillations in dual updates) through sophisticated iterative heuristics. However, these techniques remain *reactive* relying on mathematical rules to detect and resolve conflicts during the search trajectory. In contrast, our Neural Dual Warm-Start is *predictive*: it utilizes global data-driven features to anticipate contention and initialize the duals in a conflict-free state, effectively bypassing the price wars entirely. While we benchmark against LAPJV (the standard backend in scientific computing libraries like `SciPy`), our learned duals are theoretically compatible with any dual-based solver, including Auction algorithms.

## M.2. Learning-Augmented Optimization Frameworks

Recent advances in learning-augmented optimization have highlighted the potential of integrating predictive models with classical algorithmic frameworks to improve both efficiency and robustness. Foundational work on *algorithms with predictions* established a general theoretical framework for analyzing algorithms that leverage imperfect predictive information, outlining design principles and competitive trade-offs between accuracy and robustness (Mitzenmacher & Vassilvitskii, 2022). Building on this foundation, Khodak et al. (2022) has investigated how predictive models can be learned jointly with algorithmic decision processes, bridging online learning and prediction-driven algorithm design. Within combinatorial settings, Dinitz et al. (2022) have demonstrated the effectiveness of combining diverse predictive strategies to achieve strong performance guarantees. The incorporation of learned side information has further enhanced classical discrete optimization methods, such as *seeded graph matching* using graph neural networks to improve structured matching performance (Yu et al., 2023). Furthermore, Antoniadis et al. (2025) provides refined bounds integrating learned information into combinatorial optimization. In addition, the work on *binary search with distributional predictions* has studied how distributional side information can guide adaptive search procedures, producing near-optimal trade-offs between robustness and efficiency (Dinitz et al., 2024).

Beyond combinatorial problems, related efforts in continuous optimization have investigated how learning-augmented and data-driven approaches can accelerate convex and linear programming methods. For example, Sakaue & Oki (2023) proposed learning-based initialization techniques for convex minimization, showing that predictions close to optimal solution sets can substantially reduce optimization time. More recently, theoretical studies have examined the expressive power of graph neural networks for solving general Linear Programming, where Chen et al. (2023) proved that GNNs can approximate LP solutions, and Qian et al. (2024) demonstrated that Message Passing Neural Networks can emulate Interior Point Methods for continuous Linear Programming.

## M.3. Additional Neural Solver Details

Specific neural architectures like GLAN (Liu et al., 2024) and MAGNET (Loveland et al., 2025) rely on constructing a graph where edges represent potential assignments. For a dense cost matrix, this graph has $N^2$ edges. Message-passing on this graph incurs a memory cost of $\mathcal{O}(N^2 \cdot F)$, where $F$ is the feature dimension (for $N = 16,000$, $N^2 = 2.56 \times 10^8$). Even with minimal features, storing the computation graph for backpropagation exceeds the VRAM of modern GPUs. This structural limitation prevents these primal-based methods from scaling to large sizes addressed in our work.

