# OpenReview forum: "Learning-Augmented Scalable Linear Assignment Problem Optimization via Neural Dual Warm-Starts"
_ICML.cc/2026/Conference — ICML 2026 regular_

### Official Review · Reviewer_WnnD · 2026-03-11

**Soundness:** 2
**Presentation:** 3
**Significance:** 3
**Originality:** 3
**Overall Recommendation:** 4
**Confidence:** 3

**Summary:**

This paper studies learning-augmented exact optimization for the LAP. Instead of predicting a discrete matching directly, the approach predicts dual variables / potentials and performs a dual warm-start for an exact solver (notably Jonker–Volgenant (LAPJV)). A key technical device is the Min-Trick, which constructs column potentials from predicted row potentials to ensure dual feasibility of the warm-start. To maintain worst-case safety, the paper introduces a fallback mechanism based on an equality subgraph density criterion, intended to revert to a safe initialization when the learned warm-start is likely harmful. Architecturally, the paper proposes RowDualNet to avoid the $O(N^2)$ memory footprint of dense GNN-style models, emphasizing linear memory and a Top-K refine step. The paper reports consistent runtime gains on synthetic and real datasets while preserving exact optimality via the underlying solver.

**Compliance With Llm Reviewing Policy:**

Affirmed.

**Final Justification:**

The rebuttal clarifies GPU execution details (chunked B×N blocks in PyTorch), provides gauge-fixing for dual labels, and adds sensitivity evidence supporting the equality subgraph density criterion and permutation robustness. These address several of my main concerns and move my assessment upward.

**Key Questions For Authors:**

1. Please clarify whether Min-Trick and equality subgraph density are computed on CPU or GPU, and provide implementation details (kernel strategy, bandwidth considerations, matrix storage format). If the matrix is implicit or generated on the fly, how do you reconcile that with full scans?
2. Since dual solutions can be non-unique, what exactly does LAPJV output as the training label for RowDualNet? Do you apply any gauge fixing / normalization for potentials? How stable are these labels across LAPJV implementations and numerical settings?
3. Provide quantitative evidence that the density criterion predicts the number of augmentations / runtime. Do you have counterexamples where density is high but the learned warm-start is still harmful? Conversely, are there benign cases misclassified as unsafe?
4. Have you compared against stronger classic initializations or other learned-dual warm start methods in the regime where they run? Can you demonstrate that the approach is not specific to LAPJV by testing at least one additional exact solver that supports dual initialization?
5. How sensitive are results to Top-K refine choice (K vs N scaling)? Also, the paper mentions using positional encodings / features; how do you ensure this does not introduce harmful index bias when row/column ordering has no semantic meaning?

**Limitations:**

1. Even if memory is reduced via RowDualNet (linear memory), Min-Trick and equality subgraph density appear to require scanning the full cost matrix, which may be infeasible in settings with implicit costs or strict online constraints.
2. The method depends on LAPJV-generated dual labels. This supervision is expensive and may encode solver-specific tie-breaking or numerical artifacts, potentially limiting transfer.
3. Since fallback “almost never triggers” in the reported experiments, the paper does not convincingly demonstrate that the safeguard prevents degradation on truly adversarial or pathological instances.

**Strengths And Weaknesses:**

Strengths:

1. The narrative is coherent: the learned component provides advice via dual warm start while LAPJV still guarantees optimality.
2. The Min-Trick provides a simple, solver-compatible way to guarantee a dual feasible warm start rather than relying on iterative projection/repair of infeasible predictions.
3. The RowDualNet design explicitly targets the memory bottleneck of dense methods by staying linear memory, combined with a Top-K refine mechanism.
4. The inclusion of a fallback policy based on an equality subgraph density indicator shows awareness of the typical requirements for learning-augmented algorithms.

Weaknesses:

1. While the paper emphasizes avoiding $O(N^2)$memory, both the Min-Trick and the equality subgraph density computation appear to require scanning the full cost matrix (and the paper also mentions feature extraction with worst-case $O(N^2 \log N)$). For the largest regimes (e.g., $N=16384$), this can be dominated by memory bandwidth / implementation details. The current presentation does not provide enough low-level clarity to support strong claims about end-to-end scalability across hardware setups.
2. The paper states that fallback never triggers on many benchmarks, which raises questions about whether the density criterion is overly conservative, whether it meaningfully detects “bad seeds,” and how robust it is against false negatives.
3. Training RowDualNet to regress to LAPJV-produced duals may entangle the method with solver-specific internal choices: dual solutions may be non-unique, and different LAPJV variants / tie-breaking / numerical details could induce label drift. This raises concerns about transfer to other exact solvers and about how much of the gain is solver-mechanism-specific rather than LAP-structure-general.

---

> ### Author Rebuttal · Authors · 2026-03-31
>
> We appreciate the reviewer’s careful evaluation and the constructive points raised, which we address below. Note that Figures of charts supporting our response are provided in the [link](https://figshare.com/s/6972710a5733e62cf72b).
>
> [W1]+[L1] Every solution for the LAP must scan the matrix at least once. Since there are $\Theta(N^2)$ edges in the graph, it is information-theoretically impossible to avoid the $\Omega(N^2)$ computational complexity without making any prior assumptions.
> Our claim regarding memory efficiency refers specifically to the input of the neural network: by utilizing row-based feature extraction, we reduce the data dimensionality sent to RowDualNet from $N^2$ to $d \times N$. While exact solvers for unstructured LAP problems typically exhibit $O(N^3)$ complexity, our goal is not to improve the theoretical bound, but to provide a substantial practical speedup through a high-quality learned warm start that reduces the overall iterations required for convergence.
>
> [W2]+[L3] We can show the density criterion is meaningful. To demonstrate this, we conducted sensitivity analysis by disabling fallback and perturbing dual predictions with Gaussian noise (see icml_fallback_results.pdf in link). Results show a direct correlation: as noise increases, density metric drops as solver runtimes rise. This confirms the density criterion is a responsive indicator of prediction quality rather than a conservative threshold. The rare fallback triggers in our benchmarks indicate high-quality initial dual predictions, not lack of sensitivity. We will incorporate these results into the revised manuscript.
>
> [W3]+[L2] Our method is solver-agnostic. We use gauge-fixing during training to ensure consistent labels independent of solver tie-breaking rules. RowDualNet learns structural properties of the LAP rather than solver-specific artifacts. Unfortunately, SciPy doesn't allow injection of dual variables, thus restricting our method. We welcome insights into mechanisms the reviewer believes might cause this issue.
>
> [KQ1] Both the Min-Trick and the equality sub-graph density checks are computed directly on the GPU (to avoid extra CPU-GPU transfers). Regarding matrix storage, explicit matrices are stored in PyTorch tensors. We use PyTorch's kernels (we didn't create custom kernels). For bandwidth considerations, we compute in block chunks (BxN) and compute per block before discarding and loading the next chunk.
>
> [KQ2] Continuing from [W3], to ensure label consistency despite the non-uniqueness of dual solutions, we apply a gauge-fixing normalization to the row potentials output by LAPJV before using them as training labels for RowDualNet. This process eliminates the inherent additive constant ambiguity. We have found these labels to be highly stable across different numerical settings and standard LAPJV implementations, any minor implementation-specific variances can be easily mitigated through standard hyperparameter tuning.
>
> [KQ3] The test provided for [W2] provides quantitative evidence that the density criterion accurately predicts the runtime. Regarding counterexamples, in the case of a binary matrix (or a matrix where all indices are the same constant), the dual landscape is flat, resulting in many reduced edges that bypass the density threshold. These matrices are not common in real-world scenarios, and even then, in these cases, a greedy algorithm performs best (and faster than standard executions).
>
> [KQ4] As we stated in [W3], our approach is independent of the specific solver implementation.  The dual potentials generated by our model reflect the mathematical structure of the problem itself, meaning the performance gains are not tied to a specific LAPJV implementation but rather to the quality of the warm-start potentials provided to the solver's dual-based initialization phase.
> Unfortunately, SciPy is designed designed as 'black box' that doesn’t allow injecting duals into its implementation
>
> [KQ5] Our empirical results demonstrate that RowDualNet is robust to the choice of the K hyperparameter. We evaluated performance across a range of K values and found that performance remains stable when K is a small constant. By contrast, scaling $K \to N$ diminishes the total speedup due to the $O(N \log N)$ overhead of full row sorting, without providing significant gains in dual quality (see icml_topk_sensitivity.pdf in link). Regarding index bias, the role of the positional encoding features is to break symmetry in "extreme" cases. To verify that the model maintains permutation invariance, we conducted a sensitivity test where the same cost matrix was evaluated across 10 random row-wise permutations. The resulting running time of the permutation instances are stable (σ≤0.24) in all cases (see icml_permutation_invariance.pdf in link), confirming that the positional features do not introduce harmful spatial or ordering bias.
>
> We commit to adding these new empirical evaluations to the final paper and appendix.

---

> > ### Author Rebuttal · Reviewer_WnnD · 2026-04-02
> >
> > The author’s reply has cleared up most of my doubts; I will consider increasing my score.

---

### Official Review · Reviewer_1b5W · 2026-03-12

**Soundness:** 2
**Presentation:** 2
**Significance:** 2
**Originality:** 2
**Overall Recommendation:** 3
**Confidence:** 3

**Summary:**

The paper studies the Linear Assignment Problem through a learning-augmented exact optimization framework. It proposes a hybrid method that uses a neural model, RowDualNet, to predict row dual variables from row-centric features, then applies a Min-Trick to construct dual-feasible column variables and warm-start an exact LAPJV solver. A fallback mechanism discards unreliable predictions and reverts to standard cold-start initialization. The main goal is to reduce the total time required to solve the assignment problem exactly while preserving the guarantees of the classical solver. Experiments are conducted against two standard exact baselines, SciPy and LAP, which shows end-to-end speedups and improved runtime stability on large-scale instances.

**Compliance With Llm Reviewing Policy:**

Affirmed.

**Key Questions For Authors:**

1.	Novelty relative to prior learning-augmented / learned-dual LAP methods. The paper positions itself within the learning-augmented exact optimization literature and also cites prior work on learned duals for assignment-type problems. Could the authors clarify more precisely what is algorithmically new relative to the closest prior learning-augmented scalable LAP methods, beyond the specific engineering combination used here? In particular, which component should be viewed as the main conceptual novelty? A clear answer would help to better assess the paper’s originality.
2.	Comparison to the closest prior learning-augmented baselines. The experiments compare against strong cold-start exact solvers, but not against prior learning-augmented or learned-dual LAP approaches. Could the authors comment on whether such methods were implemented or considered, and if not, whether they could provide even a limited-scale head-to-head comparison on moderate-size instances? This would materially affect my evaluation, because it is important not only for empirical completeness but also for establishing the paper’s novelty relative to the closest literature.
3.	The method relies on a set of row-centric features. Could the authors clarify how sensitive the results are to the choice of these features? For example, have the authors evaluated whether performance degrades significantly when using a reduced feature set, alternative feature groups, or simpler statistics? This would help clarify whether the gains mainly come from the proposed learning framework itself or depend heavily on feature engineering.

**Limitations:**

yes

**Strengths And Weaknesses:**

Strengths：

The method is simple and well motivated. The paper presents a clear learning-augmented design: predict row duals with RowDualNet, construct dual-feasible column duals via the Min-Trick, and warm-start the exact LAPJV solver. The overall pipeline is easy to follow, and each component has a clear functional role in the framework.

Weaknesses：

A notable weakness is the lack of comparison with prior learning-augmented or learned-dual methods for LAP. The experiments only benchmark against strong cold-start exact solvers, which shows improvement over standard baselines but does not establish how the proposed method compares to the closest existing literature. Including head-to-head comparisons, even at moderate scales, would make the empirical claims much stronger. This is also important for establishing novelty, as comparison to the closest prior learning-augmented methods is key to demonstrating the method’s distinct contribution beyond standard exact-solver baselines.

---

> ### Author Rebuttal · Authors · 2026-03-31
>
> We thank the reviewer for emphasizing that the comparison to prior work (presented in the Appendix) was not sufficiently demonstrated. We address this concern, along with the sensitivity of the results to the choice of features, as follows. Note that Figures of charts supporting our response are provided in the [link](https://figshare.com/s/6972710a5733e62cf72b).
>
> [W]
> We compare against the state-of-the-art learning-augmented method of Dinitz et al. (2021), and we are not aware of any other methods in this setting. This comparison is currently presented in Appendix G and shows major improvements in large instances. In the final paper, we will clarify this point by incorporating the most relevant results to the main body of the paper.
>
> [KQ1]
> Following response to [W], Dinitz et al. (2021) is considered the current state-of-the-art for solving via learned duals. Their approach learns "distribution-level" dual variables from the training data and therefore is not adaptive to individual problem instances at test time.
> In contrast, our primary conceptual novelty is a shift to “instance-level" (feature-driven) dual prediction, enabling the method to adapt to each input graph and thereby improving generalization. Additionally, current methods may produce infeasible dual solutions (i.e., violating u_i+v_j≤C_ij), requiring iterative correction procedures that incur additional computational overhead. Our method guarantees feasibility by construction (via the Min-Trick), eliminating the need for such post-processing and improving efficiency. Furthermore, prior empirical evaluations are limited to matrices of size up to NXN=1,000X1000, whereas we demonstrate scalability for instances where N is 16x larger, significantly extending the range of problem sizes previously considered in the literature.
>
> [KQ2]
> As noted in our answer in response to [W], in Appendix G we compared our method to Dinitz et al. (2021). Our results show consistent runtime improvements over this baseline. In particular, for matrices of size $1000 \times 1000$, their method achieves max speedups of only 1.25x in the best case, whereas our method attains speedups exceeding 1.25x even in the worst case, and surpasses 2x in large instances of 16kx16k matrices.
>
> [KQ3]
> The exact number of features is not crucial, as long as it is a relatively small constant $d$, that does not scale with the problem dimension $N$. This is a key aspect of our method, which significantly reduces the input matrix dimension from $N^2$ to $dN$.
> Naturally, fine-tuning the number of features leads to a trade-off between the number of features and coverage. To exemplify this, we trained our network with different number of features. Once with just 4 simple distributional statistics (distribution features - row min, row max, row mean and row std) and once with 13 features, including all of our chosen features, excluding positional encoding. Consequently, we evaluated the speedup on our synthetic datasets, namely dense and block-structured models, for $N=4096$ (see icml_feature_ablation_4096.pdf in link below). For the dense model, in most cases our full (21) features set and the 13 features variants build resulted in similar speedups, while consistently outperforming the simple statistics 4-feature version. The positional encoding additional feature provides gains in  more challenging settings of block-structured matrices, where finer discrimination between rows is required. Overall, the choice of d=21 represents an empirical sweet spot, balancing expressiveness and efficiency, while enabling robust performance even on difficult instances with a relatively small $d$.
> [link](https://figshare.com/s/6972710a5733e62cf72b?file=63296821)

---

> > ### Author Rebuttal · Reviewer_1b5W · 2026-04-04
> >
> > Thanks for the detailed rebuttal. I appreciate the authors’ effort to clarify the comparison to prior work and to provide additional discussion on feature sensitivity.
> >
> > Regarding the comparison to prior learning-augmented methods, I acknowledge that the authors have included a comparison to Dinitz et al. (2021) in the appendix and plan to move it to the main paper. This is a positive step.
> >
> > On the question of novelty, the clarification about “instance-level” versus “distribution-level” dual prediction is helpful. The feasibility-by-construction property via the Min-Trick is also a meaningful design choice. That said, the overall contribution still appears to be a combination of known ideas (learned duals, feasibility correction, and warm-starting exact solvers), and it remains somewhat unclear whether the conceptual advance is sufficiently distinct beyond this integration. I believe the paper would benefit from a sharper articulation of what fundamentally changes algorithmically compared to prior learning-augmented LAP approaches.

---

> > > ### Author Response · Authors · 2026-04-05
> > >
> > > We thank the reviewer again for mentioning the need to emphasize our novelty.
> > > The key novelty of our approach lies in how existing ideas are combined to construct a warm-start approach predicting a nearly optimal dual solution that directly reduces the search space while strictly maintaining optimality. Compared to Dinitz et al. (2021) prior learned-dual method, the main innovations of RowDualNet framework are:
> > >
> > > 1. Instance-level dual prediction – As previously noted, Dinitz et al approach is based on a "distribution-level" prediction, lacking adaptability to specific problem instances during testing. In contrast, we provide a conceptual innovation that involves transitioning to "instance-level" (feature-driven) dual prediction, allowing the approach to adjust to each input graph, thus enhancing generalization.
> > > 2.	The Min-Trick feasibility guarantee – Our version of the Min-Trick constructive prediction *guarantees feasibility* for any arbitrary neural output by design. This guarantee is a novel feature that is unique to our framework (as other methods require feasibility correction), which decouples the learning objective (predicting good row duals) from the combinatorial constraint (ensuring bipartite feasibility). This feature allows the significant achieved speedup.
> > > 3.	Memory footprint reduction - Our approach processes rows independently resulting in a linear $O(N·D)$ memory demand, thus improving the $O(N^2)$ memory bottleneck of traditional approaches and increasing our framework scalability. Indeed, our tested benchmarks scale to $N=16,384$, which is a $16×$fold increase over the largest instances considered in prior learned-dual work. Consequently, our novel approach eliminates a critical bottleneck that prevents prior approaches from scaling.
> > >
> > > The contribution is therefore best understood not as an integration of known components, but as an intelligent combination of employed features enabling scalability, speedup, and generalization. We will emphasize these contributions in the main paper.

---

### Official Review · Reviewer_RAa3 · 2026-03-14

**Soundness:** 4
**Presentation:** 4
**Significance:** 3
**Originality:** 3
**Overall Recommendation:** 5
**Confidence:** 4

**Summary:**

The paper considers the Linear Assignment Problem (LAP), a fundamental and practical Combinatorial Optimization problem. Given a $N\times N$ cost matrix $C$ the goal is to find a permutation matrix $P$ such that $C\cdot P$ is minimized.

The goal is to design a pipeline that, most importantly, preserves the guarantee of finding an optimal solution while also being computationally fast and scalable to large instances (in the context of this paper, a large instance means roughly 16k). LAP is usually solved with the LAPJV algorithm, which can be slow. The idea here is to learn a warm start for LAPJV so that it drastically reduces its search space.

To this end, they propose a supervised training model that learns the dual variables for the row indices. Based on these, and using the Min Trick, they derive the dual variables for the column indices. The Min Trick is essentially what enables them to obtain a feasible dual solution, which is then passed to LAPJV as a warm start. Having a feasible dual solution is exactly what guarantees the optimality of their pipeline.

**Compliance With Llm Reviewing Policy:**

Affirmed.

**Key Questions For Authors:**

- Lines 59–60: Could you clarify why a perfect matching exists in the equality subgraph?
- Proposition 1: Please clarify and justify the statement that “initializing it with a feasible dual solution is mathematically equivalent to pausing a standard execution and resuming it.” Is this equivalent to saying that, for every feasible dual solution, there exists a standard execution with an intermediate step corresponding to that feasible dual solution?
- For the ablation study, “Is Deep Learning Necessary?”, could you comment on a purely gradient-descent-based optimization method using the same objective? This seems like a simple and meaningful baseline to compare against, and it could further justify the use of a neural network.
- Lines 317–320: Could you elaborate on why the speedup factor differs between synthetic data, where it is clearly greater than 2 for large instances, and real-world data, where it struggles to approach 2? Since inference is only a forward pass (i.e., matrix multiplication), one might expect sparser matrices to lead to faster inference.
- I am also wondering why there is no comparison with ILP solvers such as Gurobi or CPLEX. Is this because such solvers may not be optimal for large instances ?

**Limitations:**

yes

**Strengths And Weaknesses:**

### Strength:
The paper is very clearly written and enjoyable to read. It studies an important combinatorial optimization problem, and one of its most appealing aspects is that the proposed pipeline is exact and does not sacrifice optimality. The experimental section is also well presented, and I appreciated that all features were listed in the appendix. Their experimental results are relatively strong-ish. Another strength is the method’s ability to generalize: although it is trained only on synthetic datasets, it still performs well on real datasets with very different distributions, such as shortest-path distance matrices. However the speed up factor is less impressive here.

### Weakness:
One draw back of their approach is that it does not remove the dependency on LAPJV. As observed in Table 1, almost 90 % of the wall-clock time is still spent on seeded LAPJV. This is understandable to some extend if someone wants to keep optimality and depend on LAPVJ. The speed up factor for real-world data is not that strong.

---

> ### Author Rebuttal · Authors · 2026-03-31
>
> Thank you for your supportive review of our work. Below are the answers for your insightful questions. Note that Figures of charts supporting our response are provided in the [link](https://figshare.com/s/6972710a5733e62cf72b).
>
> [W] It is common for many optimization algorithms that only a small fraction is devoted to initialization. Our method improves this initialization to reduce the total run time.
> On real-world datasets, we observe speedups of approximately 1.25x for Multiple Object Tracking and 1.5x for Urban Transportation Networks. While these gains are lower than the ~2x improvements observed on synthetic instances, we believe that such improvements are nevertheless significant, given that the LAP is a central combinatorial optimization problem, which has been extensively studied and its solutions were highly optimized over decades.
>
> [KQ1] A perfect matching is guaranteed to exist in the equality subgraph by the fundamental theorem of Complementary Slackness (CS) in Linear Programming. For the LAP, let X* be the optimal assignment matrix and let (u*,v* ) be the optimal dual row and column potentials.  The CS condition dictates that for every edge (C_ij - u_i* - v_j*) x_ij* = 0. Because X* is a perfect matching, x_ij*=1  for all matched edges. Therefore, the term (C_ij-u_i*-v_j* ) must exactly equal 0 meaning C_ij=u_i*+v_j*. Thus, all edges comprising the optimal matching strictly reside within the optimal dual equality subgraph.
>
> [KQ2] Our claim is not that every feasible dual solution corresponds to an identical intermediate state of some standard execution trace. Rather, the equivalence we refer to is from the solver’s operational perspective: the algorithm only requires a feasible dual solution to proceed. Consequently, initializing the solver with any valid dual potentials allows it to continue with the augmenting path phases exactly as it would after reaching such a state through its own internal computations. In this sense, the initialization is mathematically equivalent to resuming execution from a valid intermediate state.
>
> [KQ3] The same question was raised by the first reviewer 2rim answered in [W4].
> The sub-gradient heuristic is not as effective as our neural based solution. As requested, we compared the speedup of a sub-gradient ascent dual initialization with our solution for matrices of size $8192x8192$. The results show that, for an equal dual prediction running time, on average, our method achieved 2.5X speedup over a cold start, whereas the sub-gradient method only achieves a 1.5X speedup. (see icml_subgradient_ablation.pdf in the link below)
> [link](https://figshare.com/s/6972710a5733e62cf72b?file=63296830)
>
> [KQ4] For instances with sparse cost matrices, such as those arising in Multiple Object Tracking (MOT), classical solvers are already highly efficient, as the number of feasible augmenting paths to explore is significantly reduced. Consequently, the available room for improvement is inherently limited, leading to smaller relative speedups. We further emphasize that our results on real-world datasets are obtained without training on such data (i.e., in a zero-shot setting). Despite this, our method still achieves consistent speedups, demonstrating its ability to generalize and provide performance gains even without access to domain-specific, and often costly, real-world training data.
>
> [KQ5] The ILP formulation of LAP is in fact tractable, because the constraint matrix is totally unimodular. As a consequence, the LP relaxation always admits an integral optimal solution, meaning that solving the LP relaxation is theoretically sufficient to recover an optimal integer solution without invoking branch-and-bound. However, despite this favourable theoretical property, general-purpose LP solvers are not computationally competitive for this problem class in practice. Specialized combinatorial algorithms (e.g., LAPJV methods) exploit the problem structure much more efficiently. Following your question, we conducted additional experiments using a Gurobi solver on a machine with 64GB of RAM for problem sizes 1024, 2048, and 4096.
> The execution times for the Gurobi solver were recorded as 3.2 and 26 seconds for the 1024 and 2048 instances, respectively. In contrast, specialized LAP solvers exhibited execution times that were two orders of magnitude faster, specifically 0.03 and 0.15 seconds for the 1024 and 2048 instances, respectively. For larger instances, there was a notable increase in memory consumption and for the case of size 4096, the solver failed to complete the task due to out-of-memory.

---

> > ### Author Rebuttal · Reviewer_RAa3 · 2026-04-03
> >
> > Thank you, I keep my score.

---

### Official Review · Reviewer_2rim · 2026-03-18

**Soundness:** 3
**Presentation:** 3
**Significance:** 2
**Originality:** 2
**Overall Recommendation:** 3
**Confidence:** 4

**Summary:**

The authors propose a neural-net based solver for the linear assignment. The key idea is for the neural net to propose a warm start for a known efficient algorithm (LAPJV) in an effort to speed it. The method is exact because algorithm maintains exactness regardless of the initial guess. To method also implements a fallback mechanism to ensure that the algorithm is in the worst case as fast as the original. The fallback mechanism leverages a sparsity heuristic to detect whether the neural network guess is in the right track and defaults to the standard algorithm behavior otherwise. The method is evaluated extensively on benchmark instances where it shows consistent speed ups over standard implementations.

**Compliance With Llm Reviewing Policy:**

Affirmed.

**Final Justification:**

After discussing with the authors, I acknowledge that the proposed method provides a speedup over the baseline with the use of a neural network. At the same time, the improvements appear to be more prominent in certain classes of instances and more modest in others. It's unclear how much room for improvement overall there is in this setting in the first place so it feels like the overall potential impact is somewhat limited to start with.
 It also remains unclear to me how things would play out with this approach on instances with more 'global structure'. The authors suggested it would be straightforward to extend it to those cases but deemed it not necessary to demonstrate their method there. Some results in this direction would've been helpful.

I will maintain my score but I don't think that this is a bad paper by any stretch so I wouldn't oppose accepting it. I just view tthis as a more borderline submission with an interesting technical approach that is tailored to a very specific problem in a specific setting so I am a skeptical about the overall importance of the contribution here.

**Key Questions For Authors:**

See weaknesses

**Limitations:**

Yes

**Strengths And Weaknesses:**

### Strengths
- The paper is clearly written and it motivates clearly its core design decisions.
- The method is exact and the neural variant appears to just strictly improve over the base algorithm.
- The paper provides several ablations and supplementary experiments including comparisons with previous work on the topic, runtime stability tests and robustness tests.




### Weaknesses
I will focus on the empirical component because I think for the most part the methodological one is sound.
- What is a bit of unclear from the paper is the range of instances over which this learned warm start will be effective. It helps that there is a fall back mechanism, but I think we are missing a more detailed characterization of where to expect this to fail. You mention geometric graphs but what else?
- To add on to the previous point, is it fair to say that this works better on dense instances? This is the impression I get from the section on the MOT benchmark. If the instances are sparser, would the speedups be noticeable? Perhaps a demonstration of whether performance degrades gracefully as sparsity increases could help.
- Suppose some instances are out of distribution. How could your approach deal with distribution shift like this? Could you finetune on small subsampled instances of some sort or is your MLP architecture fundamentally not capable of dealing with instances like the geometric ones?
- Could you test on the instances that Dinitz et al. (2021) tested on? There seem to be some large scale instances that the paper could test on as well.
- The paper argues that a neural net provides a noticeable through its ablation. I'm curious, since the dual variables that are being warmstarted come as solutions to a relatively simple optimization problem themselves (including the mintrick), could you run a few steps of some kind of simple (sub)gradient method on the dual? More broadly, I'm just curious of what plausible simple classical heuristics one could run in place of the neural net so that we get a sense of how much signal the neural net is truly picking up.

Overall, I find the overall method interesting and simple. I am open to accepting the paper but before I can recommend acceptance I would like to see the authors' answers to my comments because I think certain aspects of the method's empirical efficacy are unclear.

---

> ### Author Rebuttal · Authors · 2026-03-31
>
> We thank the reviewer for his useful feedback, raising the need for clarification of the evaluation. Please see our responses addressing the specific concerns below. Note that Figures of charts supporting our response are provided in the [link](https://figshare.com/s/6972710a5733e62cf72b).
>
> [W1]
> Our approach relies on the informativeness of the row-wise features extracted by RowDualNet. As noted, the method performance may be limited for geometric graphs with global spatial relations, because RowDualNet prioritizes local row-based features, it may be less effective on graphs where the optimal solution is dictated by high-level, global topological structures. Our fallback mechanism successfully detected and managed such instances across all our experiments. Furthermore, the model requires sufficient variance in the input data to learn effectively, and may struggle with degenerate or purely binary matrices, which offer limited signal for feature extraction. However,  these "extreme" cases are relatively rare in complex real-world applications, and in such instances, simple greedy heuristics often suffice.
>
> [W2]
> Yes, we confirm that the speedup of our method is correlated with matrix density. To show this, we ran a test starting with dense matrices (N=4096, 8192) and observed that speedup decreases as the sparsity gradually increases (see icml_sparsity_results.pdf in the link below). Though for sparse matrices, the total run time is already minimal, which naturally limits the margin for relative speedup.
> [link](https://figshare.com/s/6972710a5733e62cf72b?file=63296827)
>
> [W3]
> Our empirical results are already “out-of-distribution". We trained RowDualNet only on synthetic datasets yet tested on MOT and transportation networks. As noted in our answer for [W1], geometric graphs are not expected to be suitable for our method.
>
> [W4]
> Dinitz et al. (2021) tested three datasets. We tested their proposed "Type-Model" dataset (which we refer to as the Block-Structured Model). We didn’t test the other two datasets, KDD and Covertype, as these are geometric instances which, as noted, our method is not planned to be effective (these were generated by mapping data points into ${R}^d$ and setting edge weights to their Euclidean distances). We will make this explicit terminology mapping and the geometric exclusion criteria clearer in the main text.
>
> [W5]
> The sub-gradient heuristic is not as effective as our neural based solution. As requested, we compared the speedup of a sub-gradient ascent dual initialization with our solution for matrices of size $8192x8192$. The results show that, for an equal dual prediction running time, on the average, our method achieved 2.5x speedup over a cold start, whereas the sub-gradient method only achieves a 1.5x speedup.  (see icml_subgradient_ablation.pdf in the link below)
> [link](https://figshare.com/s/6972710a5733e62cf72b?file=63296830)

---

> > ### Author Rebuttal · Reviewer_2rim · 2026-04-04
> >
> > I appreciate the additional results. These certainly help but I still have some reservations.
> >
> > 3. What I am asking is how could you rectify that? What criterion dictates which instances are not suitable for your method and how could that be addressed. If it's more 'global structure', how would you deal with that? I wouldn't be surprised if there are other families of instances like this so I'm curious if the method could be reasonably adapted to mitigate such issues or if the method is fundamentally incapable of that.
> >
> > 5. Could you run the subgradient method for the rest of the datasets. It's important to understand how much the neural approach stacks up and where it can be truly beneficial compared to simpler alternatives.

---

> > > ### Author Response · Authors · 2026-04-05
> > >
> > > Thank you for your insightful questions.
> > >
> > > **[3]**  Our general warm-start framework is widely applicable assuming a proper selection of features, and is inherently agnostic to the specific feature design. The current implementation of RowDualNet is particularly effective for problems where dual information can be deduced from row-local statistics. In instances where the problem exhibits a global structure, then global-context features (e.g., spectral statistics of the cost matrix) could be incorporated to better capture such dependencies. The current results already demonstrate strong performance using row-local features, so augmenting them with global information is just a straightforward extension rather than a necessary validation of the approach. In general, the chosen features should reflect knowledge on the class of problems being addressed, reflecting the inductive bias necessary for any efficient machine learning (“no-free lunch theorem”). We can also recommend using domain-specific fine-tuning, by training RowDualNet with challenging domain specific instances exhibiting stronger global dependencies. However, we didn’t find it necessary for our experiments to show significant performance improvements over the baselines.
> > >
> > > **[4]** As requested, we ran the subgradient test on the rest of the datasets. The results can be viewed in the following [link](https://figshare.com/s/1ef07cf8c5070a9b8ff0). As we expected, RowDualNet shows significant performance improvement across all dataset scenarios. Note that, for the relatively sparse case of the MOT dataset, the subgradient method yields inferior results compared to the basic cold start method, likely due to the relatively rapid execution times of sparse matrices.

---

### Decision · Program_Chairs · 2026-04-30

**Decision:**

Accept (regular)

**Comment:**

The paper presents a way to warm start a linear assignment solver so that ultimately the overall process goes faster. The warm starting is done through a neural network trained on synthetic problems.
Reviewers agree on the experimental gains, i.e. significant speedups on synthetic and modest ones on real world out of training distribution problems. The conceptual contribution is incremental, in that it is a combination of existing ideas. The rebuttal is positively acknowledged. All in all this is an incremental contribution with some merit.